# Brain-state mediated modulation of inter-laminar dependencies in visual cortex

Anirban Das[1,2,3,8], Alec G. Sheffield ®[4,8], Anirvan S. Nandy ®[2,5,6,7,9] & Monika P. Jadi ®[1,2,7,9] ✉

Spatial attention is critical for recognizing behaviorally relevant objects in a cluttered environment. How the deployment of spatial attention aids the hierarchical computations of object recognition remains unclear. We investigated this in the laminar cortical network of visual area V4, an area strongly modulated by attention. We found that deployment of attention strengthened unique dependencies in neural activity across cortical layers. On the other hand, shared dependencies were reduced within the excitatory population of a layer. Surprisingly, attention strengthened unique dependencies within a laminar population. Crucially, these modulation patterns were also observed during successful behavioral outcomes that are thought to be mediated by internal brain state fluctuations. Successful behavioral outcomes were also associated with phases of reduced neural excitability, suggesting a mechanism for enhanced information transfer during optimal states. Our results suggest common computation goals of optimal sensory states that are attained by either task demands or internal fluctuations.

Adaptive information processing, comprised of local computations and their efficient routing, is crucial for flexible brain function. Attention to features, such as color and shape, or locations of interest in the visual scene is regularly deployed by the visual system to enhance object recognition[1,2]. Object recognition is mediated by neural computations distributed across the ventral visual hierarchy in the cortex - V1, V2, V4, and IT – with increasing receptive field sizes and complexity of features encoded along the way[3,4]. Deployment of spatial attention is thought to modulate these hierarchical computations in a way that aids both information representation and information transfer. Neurons that encode an attended visual stimulus increase their activity at various stages (V1, V2, V4, LIP, MT) of visual processing[5–13], with neuronal gain especially prominent in the later stages of the hierarchy. Attentional gain increase is also cell-class[14–18] and layer specific in area V4[7]. Based on pairwise correlation analysis, two additional mechanisms of attentional enhancement have been

proposed: First is an improvement in the efficacy of unique information directed from one encoding stage to another, suggested by evidence along the visual hierarchy[19–23]. Based on theoretical results that even weak correlated variability can substantially limit the encoding capacity of a neuronal pool[24], a second proposal is an improvement in the sensory information capacity of an encoding stage through a reduction in shared fluctuations[25,26] of neural activity. However, pairwise analyses capture both unique and shared components of these fluctuations, and therefore cannot disambiguate the proposed mechanisms in a multi-variate system such as the cortical network. To test these proposals, it is crucial to estimate the attentional modulation of unique information flow across and shared information within the stages of the visual hierarchy. We investigated these questions in the multi-stage laminar network of visual area V4, an area in the ventral visual stream that is strongly modulated by attention[13,27,28]. The visual cortex has a laminar organization and both sensory computations and

[1]Department of Psychiatry, Yale University, New Haven, CT 06511, USA. [2]Department of Neuroscience, Yale University, New Haven, CT 06511, USA. [3]Design and Patterning AI Group, Intel Corp., Hillsboro, Oregon 97124, USA. [4]Interdepartmental Neuroscience Program, Yale University, New Haven, CT 06511, USA. [5]Department of Psychology, Yale University, New Haven, CT 06511, USA. [6]Kavli Institute for Neuroscience, Yale University, New Haven, CT 06511, USA. [7]Wu Tsai Institute, Yale University, New Haven, CT 06511, USA. [8]These authors contributed equally: Anirban Das, Alec G. Sheffield. [9]These authors jointly supervised this work: Anirvan S. Nandy, Monika P. Jadi. ✉e-mail: monika.jadi@yale.edu

information flow patterns are layer-specific, forming the building blocks of the ventral visual hierarchy[29–33]. We hypothesized that the deployment of spatial attention strengthens unique inter-layer information transfer between and weakens shared information within the input and superficial layers, both of which are crucial nodes of feed-forward information flow along the ventral visual hierarchy[30,34]. To test this, our primary goal was to quantify unique statistical dependencies between the populations of each layer, which requires characterizing the joint spiking activity of the laminar ensemble. Using network-based statistical modeling, we estimated the strength of inter-layer information flow by measuring statistical dependencies in the V4 network that reflect how the cortical layers uniquely drive each other's neural activity. We quantified their modulation across attention conditions (attend-in vs. attend-away) in a change detection task. Using the partial information decomposition framework[35,36], we estimated the modulation of shared dependencies, specifically in the putative excitatory subpopulations. Additionally, we assessed if optimal sensory processing strategies are common to brain states that

could be either task-driven or resulting from endogenous state fluctuations[37–39].

## Results

### Information decomposition framework to distinguish the modulation of unique and shared sources of dependencies in a network

In a multi-actor system, it is important to understand how information carried by multiple source variables about a target variable is distributed over the source variables. Partial information decomposition[35,36] (PID) seeks to characterize the multivariate Shannon information that a set of source variables contains about a target variable into basic atoms of information. Depending on the nature of the underlying interactions between actors, multivariate information can be decomposed into two or more key components: uniquely directed from each source, shared by sources (Fig. 1a), or additionally, synergistically provided by two or more sources (Fig S1). Cortical neural ensembles are highly interconnected, and this can be a source

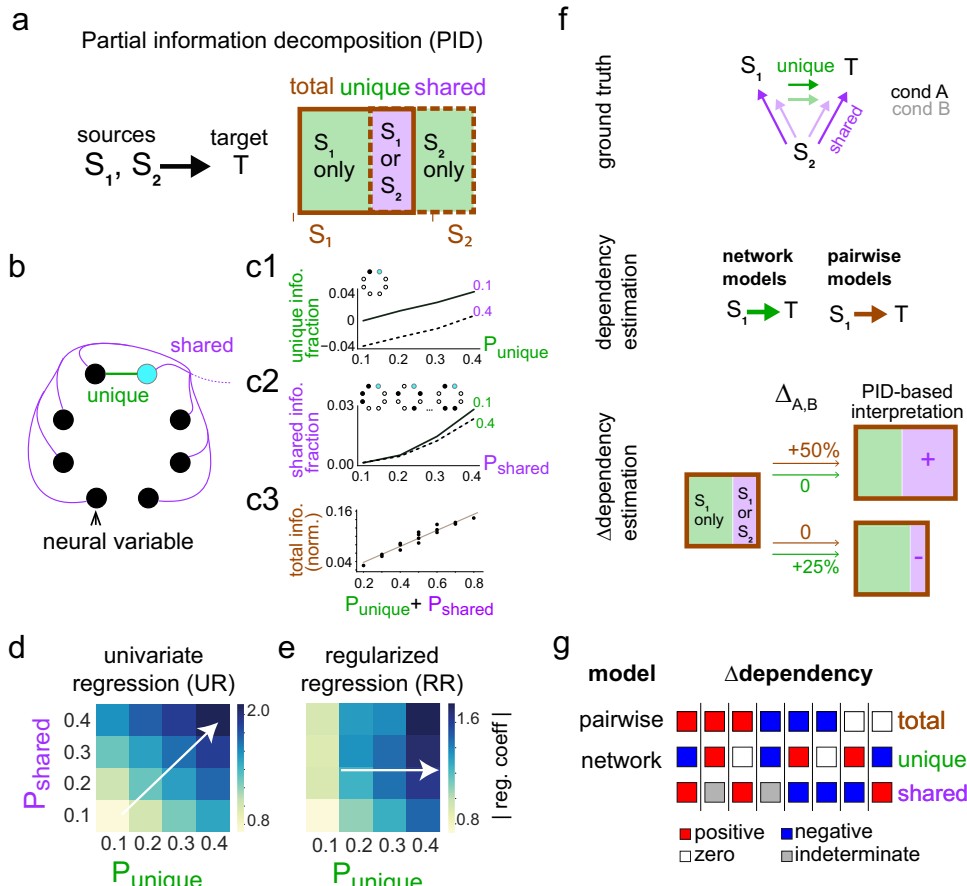

**Fig. 1 | Dependency decomposition in a multi-variate system using pairwise and network models. a** Simplified Partial information decomposition (PID) framework[35,36] based definition of types of information that multiple sources can have about a target (see Methods and Supp. Fig S1). **b** A synthetic ensemble of eight neural variables with two kinds of dependencies – unique or shared – between seven source variables (black) and one target variable (cyan). All interactions are excitatory. Strength of dependencies is determined by model parameters $P_{unique}$ and $P_{shared}$ (see "Methods"). **c1, c2** Information fraction (reduction in the proportion of total entropy) as a function of parameters ($P_{unique}$, $P_{shared}$) that control unique and shared information in the model. Information fraction estimation as a function of $P_{shared}$ (**c2**) was performed using a subset of variables in the simulated network for computational efficiency, (see Methods). **c3** Normalized total mutual information, measured by uncertainty coefficient, as a function of the sum of model parameters ($P_{unique}$, $P_{shared}$) that varied unique and shared components of

mutual information in a monotonic way. **d** Coefficients of a pairwise model (univariate logistic regression (UR)) as a function of $P_{unique}$ and $P_{shared}$. White arrow provides a visual guide for direction of highest change in coefficients. **e** Coefficients of a network model (LASSO multivariate regularized regression (RR)) as a function of $P_{unique}$ and $P_{shared}$. White arrow provides a visual guide for direction of highest change in coefficients. **f** Application of pairwise and network-based statistical models for approximate information decomposition in an example multivariate system. It illustrates interpretation of the modulation of these dependencies using the PID framework. **g** Schema for utilizing pairwise and network methods for the estimation of total (brown) and unique (green) information modulation respectively, and to infer the modulation of shared information (purple) based on the PID framework. Shaded blocks indicate indeterminate modulation direction of shared information in the network.

of both unique and shared dependencies between any pair of ensemble members (Fig. 1a, b). We hypothesized that dependencies traditionally captured by pairwise models should reflect the total information that a source has about a target, while those captured by network models should mainly reflect the unique information[40,41]. We tested this in synthetic multivariate neural data (see Methods) with parameterized unique and shared dependencies between variables (Fig. 1b, c). When we fit a univariate regression model to our data, the coefficients indeed varied as a function of the sum of shared and unique information (Fig. 1c, d). On the other hand, when a multivariate LASSO regression[42] was fit to our synthetic data, the regression coefficients were highly sensitive to the unique information and weakly so to the shared information (Fig. 1c, e). In summary, pairwise measures reflect both unique and shared sources of dependencies, whereas network-based measures that penalize correlations between source variables (such as regularized regression coefficients), mainly extract unique sources of dependencies. With a goal to identify the modulation of unique and shared dependencies across experimental conditions, we proceeded to use a combination of network-based and pairwise models, interpreted within the PID framework, to analyze the neural spiking data and decompose the modulation of unique and shared dependency components by spatial attention (Fig. 1f, g). Since neural dependencies are temporally directed and can manifest at multiple timescales, we first developed a network-based measure of Granger[43] causal dependencies that discovers unique information that a neural source has about its target in an unbiased manner.

## Unbiased estimation of Granger causal dependencies

Probabilistic graphical models (PGM)[44] capture the factorization of the joint distribution of a large number of random variables that interact with each other. The graph edges express the conditional dependence structure between the variables that form its nodes (Fig. 2a). These models provide a framework for capturing the unique dependency structures among multiple variables, without explicit assumptions about the nature of dependency. Dynamic Bayesian Networks (DBN)[45–50] is a class of PGMs wherein graph edges describe Granger causal dependencies between multiple variables in a sparse structure. The graph edges express the conditional dependence structure between the variables and their lags (Fig. 2b). Fitting our data with DBNs allows unbiased discovery of multi-timescale Granger causal[43] and cyclical dependencies, without making assumptions about their specific nature (linear/nonlinear), direction, or latency (Fig. 2c, Fig S2). In addition to fitting models with multiple lags, we estimated the probability of discovered dependencies (referred to henceforth as weights of a graph edge) and used it as a measure of their strength[51] (Fig. 2c, d). We used a time-shuffled estimate of the edge weights to determine the statistical significance of the discovered dependencies (Fig. 2c). We refer to this approach as multi-timescale weighted Dynamic Bayesian Networks (MTwDBN). To validate our analysis pipeline, we simulated a compartmentalized network of stochastically spiking excitatory and inhibitory neurons that were recurrently connected within compartments, in addition to inter-compartmental excitatory connectivity (Fig. 2e). When tested on synthetic data based on spiking activity in the network, the MTwDBN approach robustly recovered the dependency structure (Fig. 2e-g), and performed significantly better than regularized regression, especially when the populations were sparsely sampled (Fig. 2h) as is the case in neural recordings[52–54]. Additionally, MTwDBN outperformed existing unweighted[50] or fixed-threshold weighted DBN methods[51] in uncovering the dependency structure (Fig. 2h). Finally, in addition to dependency strength, edge weights in MTwDBN provided a more accurate estimate of the dependency structure (Fig. 2i). Based on these findings, we proceeded to analyze neural data within the PID framework, using the modulation of MTwDBN edge weights as a measure of

the changes in unique dependencies across attention conditions (Fig. 2j).

## Attentional modulation of unique and shared dependencies in V4 ensembles

Visual area V4 is strongly modulated by attention[15,16,25,28,55], with neurons exhibiting attention-mediated gain increases in a cell-class[14–18] and layer-specific manner[7]. However, it is not known whether and how attention modulates information flow in the laminar cortical circuit. To characterize this, we employed linear array electrodes and recorded neuronal activity from well-isolated single units simultaneously across the cortical depth in visual area V4 of macaque monkeys performing an attention-demanding orientation change detection task (Fig. 3, see "Methods"; 2 animals, 337 single units). In the main experiment, we presented a variable-length sequence of paired Gabor stimuli with different contrasts[15] (Fig. 3a), with one stimulus overlapping the receptive fields (RF) of the recorded sites. Attention was cued either to the stimulus within the neurons' receptive fields (IN) or to the one outside it (AWAY). The monkey was rewarded for detecting the change by making a saccade to one of the two stimuli that changed its orientation at a time unknown to the animal. We used current source density (CSD) analysis to identify different laminar compartments (superficial, input, and deep), and assigned isolated single units to one of the three layers[15] (Fig. 3b, see Methods). To characterize the modulation of information flow, we analyzed layer-wise pooled spiking activity of isolated single units only, as they could be classified based on their average spike shape (Fig. 3c, see "Methods").

We fit MTwDBN and logistic regression models to the pooled spiking activity of neural subpopulations defined by layers and neuronal types (Fig. 4). We first considered the ensemble of layer-wise populations (Fig. 4a) and quantified the net attentional modulation of dependencies across multiple timescales (Fig. 4b). At longer timescales ( > 60 ms lag)[33], while attention weakened pairwise dependencies estimated by the regression models, in agreement with previous findings[25,26], we found a strengthening of unique dependencies overall. The inferred modulation of shared component of dependencies showed a weakening by attention (Fig. 4b, bottom), thereby providing direct evidence for this previously hypothesized mechanism of perceptual improvement by attention[25,26]. MTwDBN-based unique dependencies between layers, specifically those between the input and superficial layers, an important link in feedforward processing, were strengthened by attention (Fig. 4c). On the other hand, shared dependencies within the input and superficial layers were inferred to be weakened by attention (Fig. 4d). Surprisingly, unique dependencies within the layers were strengthened by attention at most timescales (Fig. 4e). Taken together, these results provide direct evidence that attention improves unique dependencies both within and across stages of the laminar circuits of the ventral visual hierarchy, while weakening shared dependencies within these stages (Fig. 4f).

Neuronal cell classes in the cortex contribute differentially to information processing[15,16,56]. To test if the above results hold when we allow the discovery of cell-class specific dependencies, we next analyzed an ensemble of broad- and narrow-spiking layer-wise populations (Fig. 4g) and quantified the net attentional modulation of dependencies across different timescales (Fig. 4h). The pattern of net modulation of unique and shared dependencies in this ensemble largely mirrored that discovered in the layer-wise aggregated ensemble, specifically at the longer timescales. Same was the case for the modulation of unique dependencies between layers and shared dependencies within layers (consistent across both animals) (Fig. 4i, j). Theoretical predictions[24] regarding the effects of shared spike-count covariation on information representation apply primarily to excitatory neurons which are the primary projection neurons in the cortex. When we quantified the attention modulation of shared dependencies within layers in a cell-class specific manner, we found that the broad-

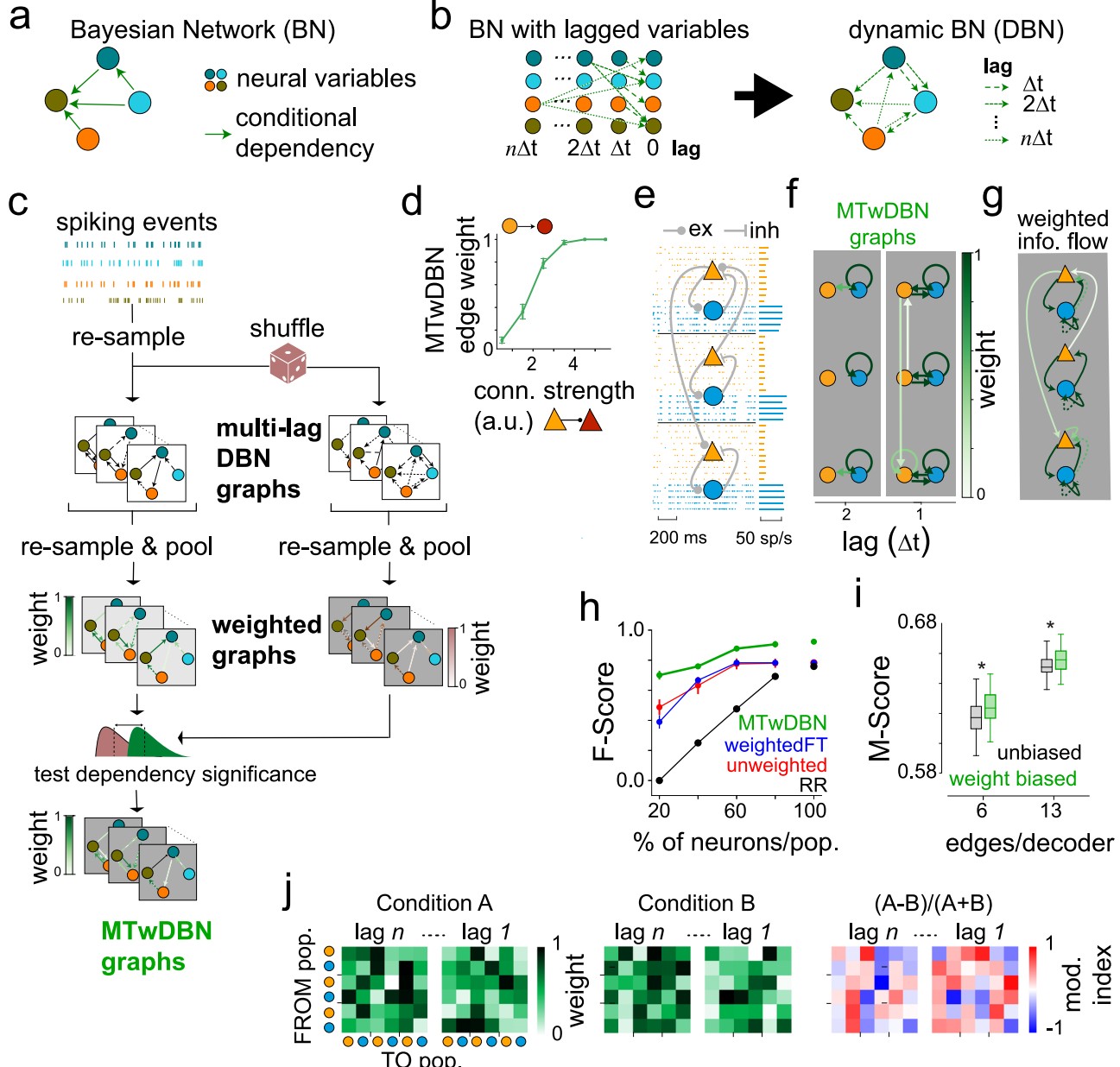

**Fig. 2 | Multi-timescale & weighted Dynamic Bayesian Networks based estimation of unique dependencies in a sparsely sampled recurrent neuronal network. a** Bayesian Networks for graph representation of dependencies in a multivariate system. **b** Dynamic Bayesian Networks (DBN) for graph representation of dependencies in multivariate time series data. **c** Analysis flow for multi-timescale weighted DBN (MTwDBN) graphical model fitting. **d** Edge weight of MTwDBN graph as a function of connection weight in a 2-population simulated network using the pipeline in **c**. Error bars indicate 95% confidence interval ($n = 100$ weighted DAGs). **e** Spiking activity of 6 subpopulations in a simulated network with recurrent connectivity. Connectivity is visualized in the overlaid schematic. **f** Directed dependencies (edge in the graph) in the simulated network in **e**, estimated using MTwDBN fitting. **g** Summary graph of dependencies across all timescales from **f**. Solid and dashed lines indicate two different timescales. **h** F-score (harmonic mean of precision and recall of dependency structure) as a function of % of neural

spiking population observed. F-score was estimated for shuffle corrected weighted DAGs (MTwDBN, green), weighted DAGs with a fixed threshold (weightedFT, blue), unweighted DAGs (red), or LASSO regression, an example of regularized regression (RR) models (black). Each point represents the average of five separate runs, except 100% (single run). Error bars indicate standard deviation, some error bars are smaller than symbol size. **i** DBN decoder accuracy with different sizes of MTwDBN DAGs. Decoders were trained to predict population activities using a subsample of shuffle-corrected edges (see "Methods"). Graph edges for the decoder were sampled from the learned structure either in an unbiased fashion (black) or biased with the edge weights (green). Box indicates lower quartile, median, and upper quartile; whiskers indicate range of data points ($n = 100$ model seeds). Asterisk (*) indicates significant differences between unbiased and weight biased M-Scores ($p < 0.001$, two-tailed paired t-test, Bonferroni adjusted). **j** Schema for estimating modulation of unique dependencies in a network of neural populations, using MTwDBN.

spiking population (putative excitatory neurons) showed a robust weakening of shared dependencies within layers (consistent across subjects) (Fig. 4j). On the other hand, the narrow-spiking population (putative inhibitory neurons) showed a distinct pattern, one that was dominated by a strengthening of shared dependencies within layers. Consistent with the findings in the aggregated ensemble, unique

dependencies within the layers were strengthened by attention at most timescales (Fig. 4k). Taken together, these results provide direct evidence that attention specifically weakens shared dependencies in the projection populations within encoding stages, in addition to robustly improving unique dependencies both within and across stages of the ventral visual hierarchy (Fig. 4l).

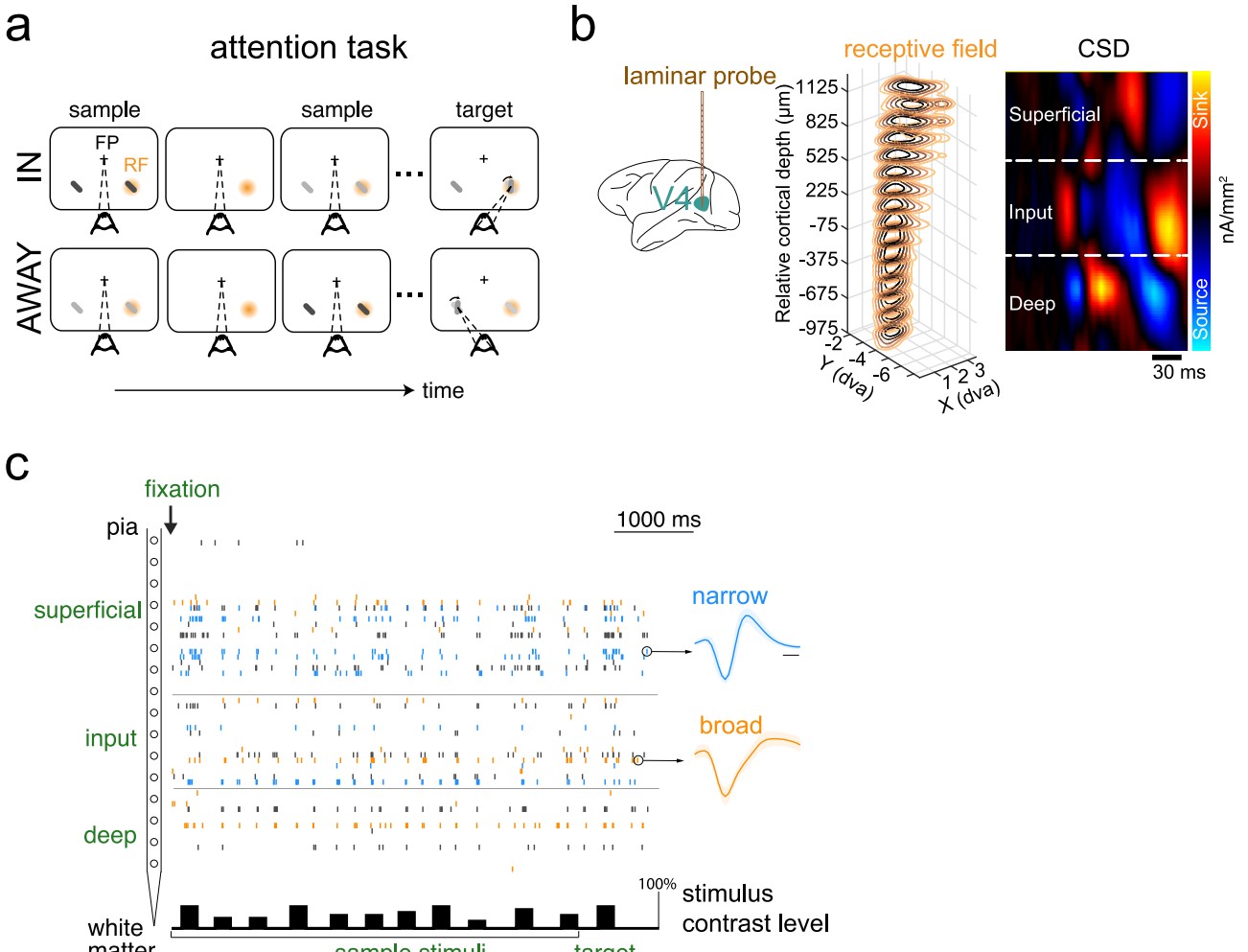

**Fig. 3 | Laminar recordings in area V4. a** Experimental protocol: Paired Gabor stimuli with varying contrasts (see "Methods"); one stimulus was presented inside the receptive fields (RFs) of the recorded neurons and the other at an equally eccentric location across the vertical meridian. Attention was cued either to the neurons' RFs (IN) or to the location in the contralateral visual hemifield (AWAY). The orientation of one of the two stimuli changed at a random time. The monkey was rewarded for detecting the change by making a saccade to the corresponding location. Task difficulty was controlled by the magnitude of orientation change. **b** Left, Recording approach: Laminar recordings in visual area V4. *Middle*, Stacked contour plot showing spatial receptive fields (RFs) along the laminar probe from an example session. Alignment of RFs indicates perpendicular penetration down a cortical column. Zero depth represents the center of the input layer as estimated with current source density (CSD) analysis. *Right*, CSD is displayed as a colored map. The x-axis represents time from stimulus onset; the y-axis represents cortical depth. The CSD map has been spatially smoothed for visualization. **c** An example trial showing single-unit activity across the cortical depth in the attend-in condition. The time axis is referenced to the appearance of the fixation spot. Spikes (vertical ticks) in each recording channel come from either single units (blue, orange) or multi-units (black). Spike waveforms for an example narrow-spiking (blue) and a broad-spiking (orange) single unit are shown. The bars at the bottom depict stimulus presentation epochs, with height indicating relative stimulus contrast. The brain schematic in (**b**) is adapted from Nandy, A.S., Nassi, J.J., Jadi, M.P., Reynolds, J.H. (2019) Optogenetically induced low-frequency correlations impair perception eLife 8:e35123. https://doi.org/10.7554/eLife.35123 and is under a CC BY license: https://creativecommons.org/licenses/by/4.0/.

## Modulation of unique and shared dependencies in V4 ensembles by behavior outcome

To test if the pattern of inter- and intra-layer dependency modulation that we observed is a signature of brain states that are optimal for perceptual behavior, we analyzed the laminar ensemble activity for a subset of trials within the attend-in condition in which the animal was equally likely to correctly detect (Hit) or fail to detect (Miss) the orientation change (Fig. 5a, b). Controlling for task and stimulus conditions, these behavioral fluctuations are thought to arise from endogenous brain state fluctuations such as attentional sampling and arousal changes[37–39,57–61]. The pattern of net modulation of unique and shared dependencies across behavioral outcomes (Fig. 5c) largely mirrored the pattern discovered across attentive states (Fig. 5d) at lags longer than 60 ms. The same was the case for modulation of unique dependencies between layers and shared dependencies within layers (Fig. 5d), suggesting that

the pattern of enhanced inter-layer unique dependencies and weakened intra-layer shared dependencies is a hallmark of brain states that are optimal for perceptual performance (Fig. 5e). Combined with prior theoretical work[24], our results suggest a conceptual model of optimal states that involves enhanced inter-layer communication and improved intra-layer information capacity that are either imposed by task demands or are attained via endogenous fluctuations in brain state.

The above model hypothesizes that endogenous fluctuations modulate the hit rate through periods of lower shared variability. To test if optimal states that are associated with hit trials are also associated with a reduction in shared correlations among the projection neurons of the input layer, we estimated the probability of presentation of successful target stimuli and the probability of spiking of input layer broad spiking units, both as a function of the phase of the ongoing cortical activity. We estimated the generalized phase of the

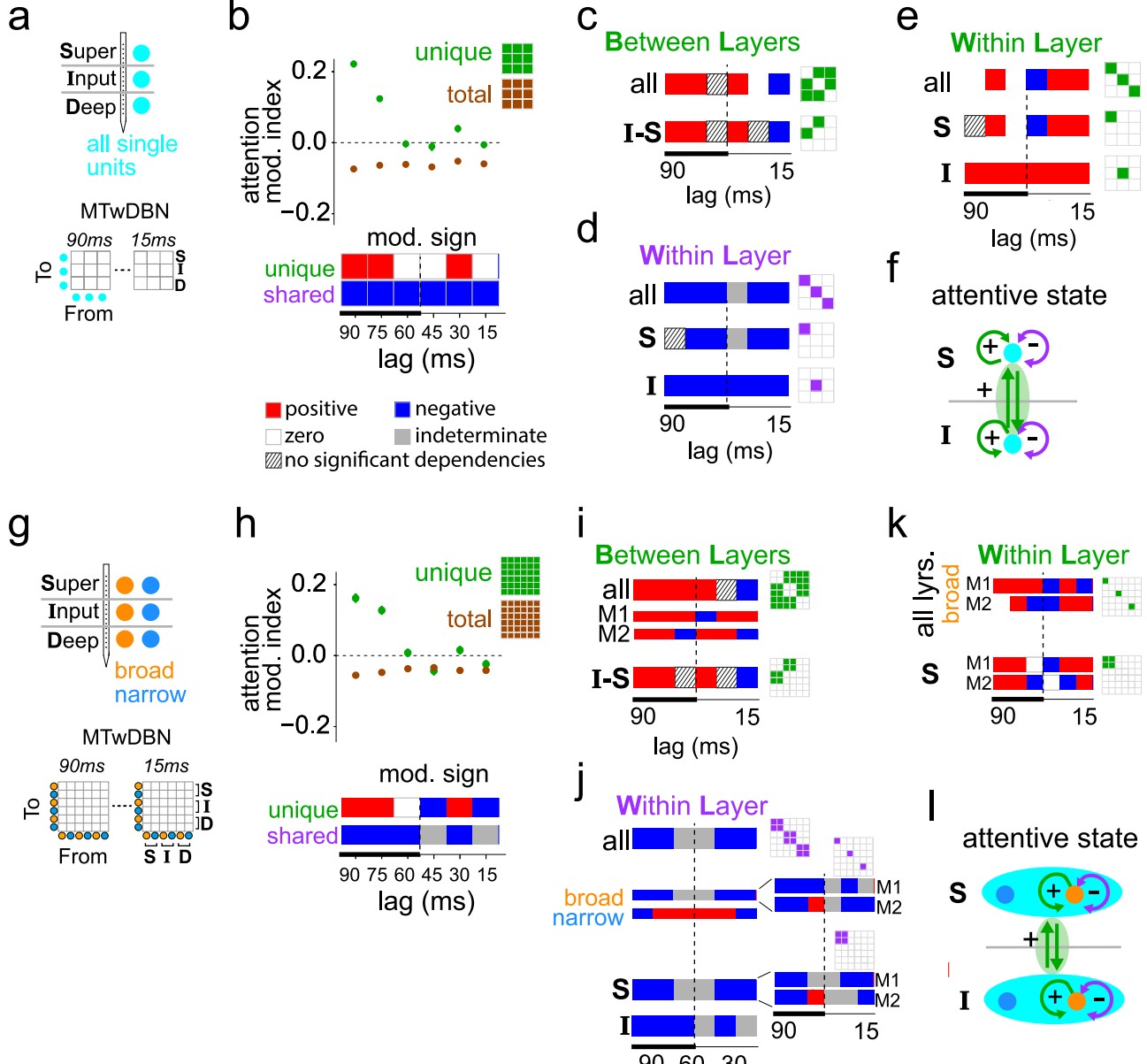

**Fig. 4 | Modulation of dependencies in a V4 laminar network across attention conditions. a** Neural populations used for fitting laminar MTwDBN. Current source density analysis identified different layers (superficial, input, deep), and isolated single units were assigned to one of these layers (see Methods). **b** Top: Average MTwDBN-based modulation (green) of all unique dependencies between the laminar populations. Modulation of the same dependencies as estimated by logistic regression (brown). Error bars indicate 95% confidence intervals. *Bottom*: Visualization of modulation sign of unique dependencies at different lags. Combining the modulation sign of unique and total dependencies (see *Top*), PID framework-based estimated modulation sign of shared dependencies (using schema in Fig. 1g) is also shown for different lags. Thicker line along the time axis indicates the timescales of attentional modulation in prior studies[38,39]. **c** Sign of average modulation of unique dependencies between layers (bi-directionally). **I**: input layer; **S**: superficial layer. **d** Sign of average modulation of shared dependencies within layers. **e** Sign of average modulation of unique dependencies within layers. **f** Summary of

dependency modulation pattern. **g** Neural populations used for fitting laminar MTwDBN. Isolated single units were classified as broad- and narrow-spiking based on peak-to-trough duration in their average spike shape (see Methods). **h** *Top*: Average MTwDBN-based modulation (green) of all unique dependencies between the cell-type specific laminar populations. Modulation of the same dependencies as estimated by logistic regression (brown). *Bottom*: Visualization of modulation sign of unique dependencies and PID framework-based estimated modulation sign of shared dependencies is also shown for different lags. **i** Sign of average modulation of unique dependencies between layers. **j** Sign of average modulation of shared dependencies within layers for all, broad or narrow populations. M1, M2: subject-wise; broad, narrow: cell-class specific. **k** Sign of average modulation of unique dependencies within layers. **l** Summary of dependency modulation pattern. See Fig S3 for modulation indices in (**c**–**e**, **i**–**k**). Data points in (**b**, **h**) indicate mean, error bars indicate 95% confidence interval (*n* = 5000 bootstraps).

band-filtered (5–40 Hz) local field potential signals in the input layer (see "Methods"), and calculated the probability of a hit-causing target onset and of neuronal spikes at different phases (Fig. 5f). We found a clear phase dependence of response onset of hit targets (Fig. 5g, top). These phases were also associated with a lower excitability of broad spiking cells (Fig. 5g, bottom), suggesting that the improved

performance in optimal states occurs during phases of lower than average spiking probability of putative excitatory neurons in the input layer. Interestingly, the excitability of superficial layer putative excitatory neurons, the primary candidates that project to downstream cortical areas in the ventral stream, was independent of the phase of the ongoing activity in the superficial layers of V4 (Fig S5).

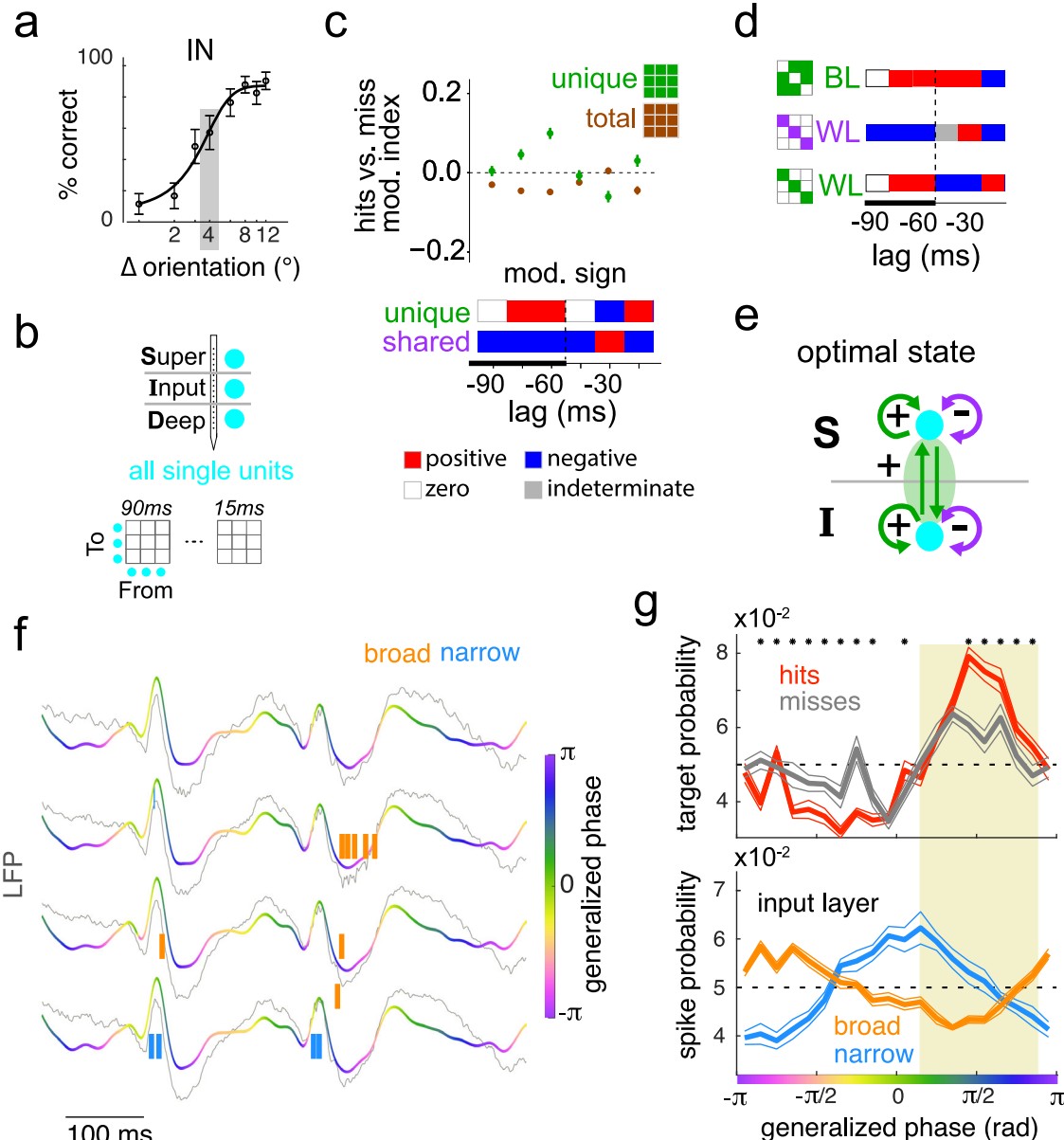

**Fig. 5 | Modulation of dependencies in a V4 laminar network across behavioral outcomes at perceptual threshold. a** Example session showing performance as a function of task difficulty. Gray box: threshold orientation change at which the animal was equally likely to correctly detect (hit) or fail to detect (miss) the change. Error bars indicate standard deviation ($n = 20$ jackknifes). **b** Laminar populations used for multi-lag analysis. **c** Modulation magnitude (top) and sign (bottom) of all unique (green) and total (brown) laminar dependencies in **b** across Hits and Misses at perceptual threshold. Estimated modulation sign of shared dependencies (bottom, see Fig. 1g). Data points indicate mean, error bars indicate 95% confidence interval ($n = 5000$ bootstraps). **d** Modulation sign of between layer (BL) and within layer (WL) dependencies. See Fig S4 for modulation indices. **e** Summary of laminar dependency modulation pattern. **f** Wideband (5–40 Hz) LFP signals (colored lines)

overlaid on the raw LFP (0–200 Hz) signals (gray) in the input layer in a portion of an example session. The generalized phase (color-coded) depicts the dominant phase of the wideband LFP. Vertical ticks indicate single-unit spikes in the corresponding channel (see "Methods"). **g** Top: Target stimulus presentation probability as a function of the generalized phase of the LFP (adjusted for cortical delay), separated by HIT and MISS trials. Asterisk (*) indicates phases with significant differences ($p < 0.05$) between the two trial types (Ranked sum test, corrected for multiple comparisons). Bottom: Spike probability in the input layer as a function of generalized phase of the LFP, separately estimated for putative excitatory (broad) and putative inhibitory (narrow) units. For other layers, see Fig S5. Error bands indicate standard error of the mean.

## Discussion

The laminar network is considered a canonical circuit that constitutes a key computational unit in the cortex. While anatomical connectivity maps have identified the key variables of this unit[30,34], access to the functional connectivity that determines the resulting computations has remained elusive. Laminar recordings in awake animals transitioning across behavioral states have allowed us to observe the neural variables that are expected to play a significant role in these computations. Using a combination of network-based dependency models and information

decomposition, we show that the dependency structure in the laminar cortical ensemble is modulated by attention in a layer and cell-type specific manner. The input and superficial layers in V4 are crucial nodes of feedforward information flow along the ventral visual hierarchy, with the input layer receiving information from earlier visual areas (V1, V2) and excitatory projection neurons in the superficial layer sending information to the next stage of this hierarchy, namely the inferotemporal cortex[30,34]. We find that unique dependencies between input and superficial layer populations are strengthened by attention, as well

as during successful behavioral outcomes within the attentive state. Our finding suggests that enhanced unique information transfer between encoding stages of the laminar hierarchy is a hallmark of behaviorally optimal sensory processing. This is in line with the observation of excitability phase alignment between layers 4 and 3 of V1 during visual attention[61]. On the other hand, shared dependencies within the putative excitatory laminar populations are weakened by attention, as well as during successful behavioral outcomes. Prior theoretical and modeling studies have proposed that a reduction in shared correlations can enhance both the information capacity[24] and signal-to-noise ratio[25,26] of a neural population. Interpreted within this framework, our finding suggests that enhanced information capacity mediated by reduced shared dependencies is another hallmark of behaviorally optimal sensory processing.

An unexpected finding of our study is the attentional enhancement of unique dependencies within the (putative) excitatory laminar populations. Prior theoretical work has shown that networks with clustered (as opposed to random) architectures can result in dynamics that lead to strong correlations within the clustered subpopulations[62]. Excitatory neurons within the superficial layers (layers 2/3) in the visual cortex have been shown to exhibit higher connection probability when the neurons receive common inputs from the input layer (layer 4) and superficial layers[63]. Since our laminar populations are tightly localized in cortical space and, given their strongly overlapping receptive fields (Fig. 3b), are recipients of highly overlapping inputs, their activity fluctuations could result in within-population unique dependencies. The observed modulations of these dependencies could reflect the attentional enhancement of effective connectivity (and hence clustering) in these populations and the resulting enhancement in neural activity fluctuations.

Additionally, we find that phases of the endogenous fluctuations that are associated with optimal target presentation (resulting in hits) are also associated with reduced excitability of broad spiking neurons, especially in the input layer. Our findings are in agreement with previous reports of rhythmic shifts of neural excitability and their entrainment to the stream of sensory inputs as key mechanisms of sensory selection[57–61,64]. Interpreting the fluctuations in excitability to be at least partly based on changing correlations due to fluctuations in shared inputs, our finding suggests an additional mechanism through which weakened shared neural activity fluctuations could improve behavioral outcome: a low excitability phase of endogenous fluctuations, which is associated with reliable encoding in the visual cortex[65].

Our findings and interpretations point to a set of key questions for future studies that will shed further light on the mechanisms of optimal states. One key question that can be addressed using our computational approach, but is currently limited by the statistical power from available datasets is: how are unique and shared dependencies between pairs of populations modulated in specific causal directions, and between specific cell classes? While the current data allowed us to answer this in a limited way, recent advances in recording techniques should allow the collection of denser neural data that, in combination with the approach we successfully demonstrated in this study, would provide further insights into the computations and mechanisms of optimal behavioral states. Another equally significant question is: How are unique and shared dependencies between superficial and deep layers of the laminar network modulated during optimal behavioral states? The communication between superficial and deep layers, specifically in the direction from the former to the latter, plays a key role in the feedback computations of the ventral visual hierarchy. Theoretical frameworks of object perception suggest computations of hierarchical Bayesian Inference, with a key role for feedback pathways[66,67]. Additionally, our group has shown previously that optogenetic induction of low-frequency fluctuations in V4 impairs performance in an attention-demanding task[68]. An important follow-up to this line of investigation would be testing the hypothesis that reduced shared dependencies in laminar populations are causal to enhanced information capacity. This will require experimental paradigms with a richer set of visual stimulation, such as complex shapes, and causal manipulation of shared neural activity fluctuations in a layer-specific manner. Combining with both regular and irregular visual stimulation could further shed light on the causal role of slow endogenous fluctuations on perception[61].

## MTwDBN analysis approach

Our approach demonstrates that the information flow structure in a neural ensemble is robustly described by Dynamic Bayesian Network modeling that is adapted to include multiple lags spanning timescales of interest. This unbiased approach allows us to dive deeper into the causal functional interactions (beyond correlations) in a multivariate system with unknown nonlinear dependencies. Further, the results demonstrate that weighing dependencies using confidence measures provides a more accurate information flow structure that can be utilized to investigate how this structure is modulated by changes in brain and behavioral states. Finally, an integration of this approach with pairwise models allows us to dissect changes in distinct components of dependencies. Since temporally directed unique dependencies suggest causality and shared ones suggest common inputs, this allows us to gain insights into the network mechanisms. In comparison, prior correlational studies have quantified net dependency modulation[19,25,26]. While these provided valuable characterization of the neural correlates of attention, the underlying network mechanisms remained hypothetical.

## Comparison to other multivariate models

While approaches such as generalized linear models (GLM) have been useful in providing improved phenomenological models of individual neural responses to sensory stimulation in early sensory circuits, such as in the retina[69], they are not ideal for dependency structure learning in the highly recurrent cortical ensemble[30]. It is possible to learn the dependency structure in a laminar network by fitting a GLM model for each of the observed populations in our data, with the remaining populations as predictors. However, the assumption of the independence of predictor variables inherent in GLMs can result in the discovery of spurious dependencies due to shared inputs, and hence, a dense and likely inaccurate structure. Since structure learning in DBNs involves determining conditional independence by solving an optimization problem that penalizes density, our approach is ideal for generating a sparse and hence interpretable unique dependency structure in multivariate data.

## Current limitations of our approach

Currently, a limitation of this class of DBN models is that it does not identify functional consequences of dependencies between populations, such as enhancement or suppression of target activity. Future work in this direction will further enhance the functional interpretability of discovered dependencies. Nonetheless, these models are effective in elucidating the structure and modulation of information flow in a multivariate system such as the laminar cortical network.

In general, the discovery of dependency structure, as well as its interpretation using the PID framework, using our approach is sensitive to the inclusion of relevant neural variables. It is indeed possible that a gross classification of subpopulations as broad and narrow in our study fuses distinct but relevant neural variables. On the other hand, our three- and six-population laminar DBNs yield consistent patterns of dependency modulations, suggesting the inclusion of relevant variables for the purposes of this study. It is important to note that while statistical dependencies discovered using our approach imply directed functional connectivity, these do not necessarily imply direct anatomical connectivity: an edge in the structure could reflect indirect anatomical connectivity. Finally, the PID framework describes a synergistic component of information in a multivariate system, such as the cortical network, that we ignore for reasons of simplification. An example of

synergistic information would be that conveyed by the coactivation of two or more neural populations that are causally connected to a target population. While it is a real possibility in the laminar cortical networks, this limitation of our approach will merit revisiting when significantly denser recordings are made available through future experiments.

In summary, network mechanisms of behaviorally optimal brain states are shared across those that are either externally induced or are a consequence of endogenous brain state fluctuations. They target efficiencies in both information transfer and representation. MTwDBN-based models provide a quantitative description of information flow patterns in an ensemble. In this study, they provide a structural description of the attention modulation of dependencies in cortical space and time in a compartmentalized network, such as the laminar cortical network. They allow us to quantify the unique contributions of activity history and network interactions to information processing in such a neural ensemble. We expect this framework to extend to neuronal ensembles in other parts of the nervous system, and to play an important role in revealing flexible information-processing principles in the brain.

## Methods

### Partial information decomposition
Information theory does not provide a complete description of the informational relationships between variables in a system composed of three or more variables[35,36,70,71]. The information $I(T;S1,S2)$ that two source variables $S1$ and $S2$ hold about a third target variable $T$ decomposes into four parts: (i) $U(T;S1|S2)$, the unique information that only $S1$ (out of $S1$ and $S2$) holds about $T$; (ii) $U(T;S2|S1)$, the unique information that only $S2$ holds about $T$; (iii) $R(T;S1,S2)$, the redundant information that both $S1$ and $S2$ hold about $T$; and (iv) $S(T;S1,S2)$, the synergistic information about $T$ that only arises from knowing both $S1$ and $S2$ (see Fig S1). The set of quantities $\{U(T;S1|S2),U(T;S2|S1),R(T;S1,S2),S(T;S1,S2)\}$ is called a partial information decomposition related as follows:

$$I(T;S1,S2) = U(T;S1|S2) + U(T;S2|S1) + S(T;S1,S2) + R(T;S1,S2) \quad (1)$$

$$I(T;S1) = U(T;S1|S2) + R(T;S1,S2) \quad (2)$$

$$I(T;S2) = U(T;S2|S1) + R(T;S1,S2) \quad (3)$$

Thus, the Partial Information Decomposition (PID)[35,36] framework characterizes the mutual information between variables by decomposing it into unique, shared, and synergistic components.

In a multi-neuronal network, unique (and synergistic) components of the mutual information between neural variables due to causal interactions (unique and synergistic) are captured by directed statistical dependencies. On the other hand, shared components of mutual information between neural variables correspond to shared neuronal inputs, and the sign of their modulation can be estimated from the modulation of unique and total mutual information (Fig. 1g).

### Information decomposition with pairwise and network models
**Generation of synthetic data.** A synthetic network was generated in which unique and shared dependencies between variables were controlled. Eight variables were randomly initialized to values of 0 or 1. One variable was assigned as the target, the other seven as sources. One source was designated as the unique source. Samples were generated in the following manner: (1) the values of all variables were set to 1 with probability $P_{shared}$ and unchanged with probability $1 - P_{shared}$ and (2) the values of the target and unique source were set to 1 with probability $P_{unique}$ and unchanged with probability $1 - P_{unique}$. This created a shared dependency between all eight variables and a single unique dependency between the target and the unique source. Two thousand (2000) samples each were generated with $P_{shared}$, $P_{unique} \in \{0.1, 0.2, 0.3, 0.4\}$.

**Information estimation.** The total normalized mutual information between the unique source and the target variable was calculated by the uncertainty coefficient:

$$U(target|source) = \frac{H(target) - H(target|source)}{H(target)} \quad (4)$$

where H is entropy. Additionally, unique information between the unique source variable and target was calculated for $P_{unique} \in \{0.1, 0.2, 0.3, 0.4\}$ and $P_{shared} \in \{0.1, 0.4\}$.

Unique information was divided by the total entropy of the target and termed unique information fraction. Likewise, redundant or shared information about the target from random subsets of three source variables (repeated 100 times) was calculated for $P_{unique} \in \{0.1, 0.4\}$ and $P_{shared} \in \{0.1, 0.2, 0.3, 0.4\}$ and divided by total entropy of the target, termed shared information fraction. Unique and redundant information were calculated using the dit (discrete information theory) python package[72] with redundancy measured by pointwise common change in surprisal[71].

**Pairwise model fitting.** Univariate logistic regression was performed between the unique source and target variable using the statsmodel python package[73] for all values of $P_{shared}$ and $P_{unique}$.

**Network model fitting.** Multivariate regression was performed between all seven source variables and the target variable using the statsmodel python package using L1 penalty[74–76].

### Multi-timelag weighted dynamic Bayesian network (MTwDBN) analysis pipeline
**Fitting dynamic Bayesian network models.** A Dynamic Bayesian Network (DBN) framework was used to learn dependencies between neural populations[50]. The pgmpy python package[77] (https://github.com/pgmpy/pgmpy) and custom-written python code was used to fit all DBN models. Each binned-and-sliced data table was first bootstrapped $B$ times. For each session/condition, a hill climb tabu-search with a history window of 7 was performed 120 times, each from a unique random starting graph, to find a suitable fitting directed acyclic graph (DAG)[44]. The Akaike information criterion (AIC) was used[78,79] as the scoring metric in the tabu-search. The variables associated with the latest time slice (0 ms) were termed the effect variables; all others were termed potential cause variables. The search was restricted to DAGs where edges can only be incident on an effect variable. Edges between effect variables were allowed as they may capture dependencies at timescales shorter than our chosen bin widths, however such edges have no causal interpretation and were excluded from further analysis. The resulting DAGs were termed unweighted, as their edges (dependencies) are described in a present/absent manner. Of the 120 starting points, only the DAG with highest AIC score was used for further analysis. This resulted in $B$ unweighted DAGs per session/condition.

**Estimation of weighted DAG.** The $B$ DAGs from each session/condition were used to estimate weighted DAGs[51]. A pool of 100 weighted DAGs were estimated by taking 100 bootstrap samples from the $B$ unweighted DAGs and averaging: the weight $\in (0,1)$ for each edge corresponds to the proportion of the $B$ unweighted DAGs where the edge was present. This resulted in 100 weighted DAGs for each session/condition.

**Testing significance of DAG edges.** To test for significance of the discovered edges, the 100 weighted DAGs were compared to 100 control DAGs that were generated in the same manner as above but with time-shuffled data. The binned-and-sliced data tables were shuffled as follows: for each row, permute the data from each of the 7 time slices which arose from the same population. Finally, permute each column (corresponding to each population/time slice combination).

To test for significance, a distribution of weights was generated by combining weights for a given edge across all sessions (conditions treated separately). The distribution of edge weights from unshuffled data were compared to the distribution of edge weights from the shuffled data using a one-sided Mann Whitney U Test. If the distribution from unshuffled data had significantly higher values than from shuffled data ($p < 0.05$), the edge was marked as significant for that condition and said to have survived time shuffling.

### MTwDBN validation

**Synthetic neural network.** Synthetic neural network models were constructed using stochastic spiking neurons[80,81]. Individual neurons in the model were treated as coupled, continuous-time, two-state (active and quiescent) Markov processes. The active state represents a neuron firing an action potential and its accompanying refractory period, whereas the quiescent states represent a neuron at rest. The transition probability for the i-th neuron to decay from active to quiescent state in time $dt$ was $P_i(active \rightarrow quiescent) = \alpha_i \partial(dt)$, where $\alpha_i$ represented the decay rate of the active state of the neuron. Parameter $\alpha_i$ sets the upper bound on firing rate of the stochastically spiking neuron, akin to a refractory period. The transition probability for the i-th neuron to change from quiescent to active state (i.e., spike) was $P_i(quiescent \rightarrow active) = \beta_i G(S_i)\partial(dt)$[80]. This caused the firing probability to be a function of the input, with $\beta_i$ as its peak value. Parameter $S_i$ was the total synaptic input to neuron i, given as $S_i(t) = N_i(t) + I_i(t)$, where $N_i$ was the net input from other neurons in the local network and $I_i$ was the net external input to the neuron. The network input was $N_i(t) = \sum_j w_{ij} A_j(t)$, where $w_{ij}$ are the weights of the synapses. The activity variable $A_j(t)$ was set to one if the jth neuron was active at time $t$ and zero otherwise. The model neurons had no intrinsic capacity to oscillate because the inter-spike interval was the sum of two independent exponential random variables with parameters $\alpha_i$ and $\beta_i G(S_i)$, respectively. The model parameters were chosen as follows: Excitatory (E) and inhibitory (I) neurons in the network were differentiated based on two model parameters: $\alpha_E = 0.075$ ms, $\alpha_I = 0.4$ ms; and $\beta_E = 1$, $\beta_I = 2$.

**Two neuron network.** The model network analyzed in Fig. 2d consisted of two excitatory neurons with a single synaptic connection from the first neuron to the second neuron. Ten such two-population networks were simulated with synaptic weights $\in (0.5, 1.0, ..., 9.0, 9.5)$. 1000 trials of spiking data each 100 ms long were simulated for each model.

**Six-population network.** Synthetic laminar network analyzed in Fig. 2e-i consisted of simulating 45 neurons in total, 15 in each cortical layer (superficial, input, deep). Each layer contained 10 excitatory and 5 inhibitory neurons, giving a total of 6 populations (3 layers x 2 neuron types). The network topology for synaptic connectivity is depicted in Fig. 2e. The weights $w_{ij}$ for synaptic connections are 1 for the inter-laminar connections, 1.5 for intra-laminar connections where the pre-synaptic unit is an E unit, and -2 for intra-laminar connections where the presynaptic unit is an I unit. 1000 trials of spiking data, each 2000 ms long, were simulated using this model.

**Preprocessing for MTwDBN analysis.** Data from single units was grouped by population in each model simulation. The multi-unit spiking activity of these populations was used for the analysis. Before aggregating activities of single units into populations, $d$ % ($d$ being a pre-selected number, see *Effect of sub-sampling in synthetic laminar model*) of single units in each population were dropped (not used for analysis) from the laminar model data. The data from each trial were discretized into 1.2 ms bins of either 0, 1, or 2 to denote no spikes, one spike, or multiple spikes in a time bin. The data were lagged 2 times to give 3 time slices (2.4, 1.2, 0 ms). The data from all trials were concatenated together to generate a single data table with 6 or 18 columns (2 or 6 populations x 3 time slices). The binned and discretized spiking activity of a single population in a single time slice was viewed as a variable in either the pairwise regression or DBN framework. Data with structure as produced by this preprocessing step are termed binned-and-sliced data tables. Alternative models with between 4-7 lags were created with no improvement in accuracy (Fig S2).

**Analysis: relation between synaptic weights and DAG edge weights in two-neuron network.** Binned-and-sliced data tables from each two-neuron model were passed separately through the MTwDBN pipeline ($B = 200$) to generate 100 weighted DAGs each. Weights were obtained by averaging across the 200 unweighted DAGs. The 100 weighted DAGs were used to compute 95% confidence intervals for these weights.

**Analysis: effect of sub-sampling on recovering ground-truth in synthetic laminar network.** To assess the effect of sub-sampling neural data on MTwDBN outputs, $d$ % ($d \in \{0, 20, 40, 60, 80\}$) of neurons in each of 6 populations were dropped from the laminar network before preprocessing (*Preprocessing Synthetic Neural Network Data*). For $d \in \{20, 40, 60, 80\}$, five iterations of this procedure were performed where a different random subset of neurons was dropped. This resulted in 21 binned-and-sliced data tables.

**DBN Models: MTwDBN, weighted with fixed threshold, unweighted.** Each data table was passed through the MTwDBN pipeline ($B = 200$), generating 200 unweighted DAGs and 100 weighted DAGs. We used bootstrap sizes of 8000 rows in the MTwDBN pipeline to simulate data scarcity of electrophysiology experiments. Edges were considered significant either by surviving time shuffling (MTwDBN) or by the average weighted edge passing a fixed threshold of 0.5 (weightedFT). For unweighted models, we used the DAG with the highest AIC score out of 200 unweighted DAGs.

**LASSO multivariate regularized regression (RR).** Each data table was used to fit a multivariate LASSO regression model (sklearn.linear_model.LASSO). For each data table, one of the 0 msec time slice columns served as the predictor variable, with all other columns serving as independent variables (six models corresponding to six 0 ms time slice columns). An edge was considered present if there was a non-zero coefficient from a 2.4 or 1.2 ms time slice variable (edges within the 0 ms time slice were not considered, analogous to DBN analyses). The regularization coefficient was fine-tuned based on the F-score of the resulting model using the $d = 0$ data table. We tested a range of regularization coefficients from 0.005 to 0.05 (linearly spaced by 0.005) and found the F-score to peak at $\alpha = 0.02$.

**F-score calculation and model comparisons.** To measure how well they conformed to ground truth connectivity, an F-score was calculated for models (MTwDBN, weightedFT, unweighted, RR) fitted to a given subsampled dataset. We use a framework[50] where edges are considered regardless of the time lag in which they appear. The F-Score is defined as: $F = \frac{2RP}{R+P}$, with $R = \frac{C}{C+M}$, $P = \frac{C}{C+I}$, and C= # connections inferred by the model which are in the ground truth, M= # connections in ground truth not inferred by DBN, I = # connections inferred by the model which are not in the ground truth. R and P refer to Recall and Precision. In addition to the eight edges indicated in Fig. 2e, self-edges were regarded as part of the ground-truth to indicate firing refractory periods. F-Scores of $d \in \{20, 40, 60, 80\}$ were compared by a mixed-model ANOVA with model (MTwDBN, weightedFT, unweighted, RR) as within-subjects variable and $d$ (20,40,60,80) as between-subjects variable (RStudio, ezANOVA, revealing significant main effects ($p < 0.001$)) of model and model x drop interactions after

sphericity correction (Greenhouse-Geisser). Models were further compared for each $d$ separately using post-hoc Tukey test with Bonferroni correction. MTwDBN is considered to outperform alternative models if it has a higher F-score for all $d$ and all $p < 0.05$.

**Analysis: effect of number of time lags on recovering ground-truth in synthetic laminar model.** To assess the impact of including different number of time lags on recovering ground-truth dependencies, the synthetic laminar data was preprocessed ($d = 0$) with # of lags $\in \{3, 4, 5, 6, 7\}$. Each binned-and-sliced data table was run independently through the MTwDBN pipeline ($B = 50$) and F-scores, Recalls, and Precisions calculated.

**Analysis: validating predictive power of edge weights in synthetic laminar model.** To validate whether the edge weights contain additional information about the population dependencies above unweighted edge weights, the binned-and-sliced data table from the synthetic laminar network ($d = 0$, no neurons dropped) was divided into two data tables row-wise, data$_{train}$ (95% of data) and data$_{test}$ (5% of data). Data$_{train}$ was bootstrapped 10 times to mimic multi-session data and separately passed through the MTwDBN pipeline ($B = 50$) to generate $10 \times 100$ weighted DAGs. The set of edges in the weighted graph that survived time-shuffling are denoted E$_{set}$. Subsampled unweighted DAGs containing $n$ edges (6 or 13) were obtained by sampling without replacement from E$_{set}$ by either sampling uniformly (UW, unweighted) or sampling using the DAG edge weights as sampling weights (W, weighted). Models UW and W were used for prediction by fitting parameters to 12,000 samples from data$_{test}$ by maximum likelihood estimation[82]. From the remaining samples in data$_{test}$ not used for parameter fitting, 4,000 samples were used for validation. For each sample in the validation set, one of the effect variables was chosen at random and both UW and W were used to predict the value of this effect variable from the cause variables. The prediction accuracies of the two models were compared to the true effect variable value using the M-score[83]. This was repeated for 100 variations of UW and W (created by varying the random seed used to sample edges), and the resulting 100 M-scores were compared using a two-tailed paired t-test with Bonferroni correction for both $n = 6$ and $n = 13$. The W model is considered to outperform UW in prediction if M$_W$ > M$_{UW}$ and $p < 0.05$.

**Attention data**

**Behavioral task.** Well-isolated single units were recorded from area V4 of two rhesus macaques during an attention-demanding orientation change detection task. The task design and the experimental procedures are described in detail in a previous study[15,84]. While the monkey maintained fixation, two oriented Gabor stimuli were flashed on for 200 ms and off for variable intervals (randomly chosen between 200 and 400 ms). The contrast of the stimulus was randomly chosen from a uniform distribution of 6 contrasts ($c = [10\%, 18\%, 26\%, 34\%, 42\%,$ and $50\%]$). One of the stimuli was located at the receptive field overlap region (Attend In) and the other at an equally eccentric location across the vertical meridian (Attend Away). At the beginning of a block of trials, we presented instruction trials where the monkey was spatially cued to the covertly attend to one of two stimulus locations. One of the two stimuli changed in orientation at an unpredictable time (minimum 1 s, maximum 5 s, mean 3 s). The monkey was rewarded for making a saccade to the location of orientation change. 95% of the orientation changes occur at the cued location, and 5% occur at the uncued location (foil trials). We observed impaired performance and slower reaction times for the foil trials, suggesting that the monkey was indeed using the spatial cue to perform the task. The difficulty of the task was controlled by changing the degree of orientation change (randomly chosen from the following: 1°, 2°, 3°, 4°, 6°, 8°, 10°, and 12°). If no change occurred before 5 s, the monkey was rewarded for holding fixation (catch trial, 13% of trials).

**Electrophysiological recording.** While the monkey was performing the attention task (Fig. 3a), we used artificial dura chambers to facilitate the insertion of 16-channel linear array electrodes (laminar probes, Plexon, Plexon V-probe) into cortical sites near the center of the prelunate gyrus. Neuronal signals were recorded, filtered, and stored using the Multichannel Acquisition Processor system (Plexon). Neuronal signals were classified as either isolated single units or multiunit clusters by the Plexon Offline Sorter program. Data is available upon request.

*Laminar boundaries*: For the data collected from linear array electrodes, we used current source density analysis to identify the superficial (Layers 1–3), input (Layer 4), and deep (Layers 5 and 6) layers of the cortex based on the second derivative of the flash-triggered LFPs[15,85]. The resulting time-varying traces of current across the depth of the cortex can be visualized as CSD maps (Fig. 3b). Red regions depict current sinks in the corresponding region of the cortical laminae, while blue regions depict current sources. The input layer was identified as the first current sink followed by a reversal to current source. The superficial and deep layers had the opposite sink-source pattern i.e. source followed by sink.

*SU* classification: Cell bodies of single units with bi-phasic action potential waveforms were assigned to the same layer in which the electrode channel was situated during recordings. Units that had tri-phasic waveforms or other shapes were excluded from analyses. Units with peak-to-trough duration greater than 225µs were classified as broad-spiking putative excitatory neurons; units with peak-to-trough duration less than 225µs were classified as narrow-spiking putative inhibitory neurons (Fig. 3c). Extracellular data were collected over 32 sessions (23 sessions in monkey A, 9 in monkey C) yielding 337 single units in total. Unit yield per session was considerably higher in monkey C than monkey A, resulting in a roughly equal contribution of both monkeys toward the population data.

**Data selection.** All analyses in this study were performed on spiking data during an interval of 60–260 ms after stimulus onset excluding orientation changes. Only single units whose spike waveforms were successfully classified as broad or narrow and for whom the layer identity could be successfully discerned were used in the analysis. There were 29 sessions which had one or more such units recorded. For layer-wise analyses, only sessions with at least one unit from each layer were included (18 sessions). For broad- and narrow-spiking layer-wise (layer-class) analyses, only sessions with at least one unit from two or more populations were included (27 sessions). Pairwise and network-based dependency analysis were performed on each session separately.

**Analysis: attention conditions.** Data from Attend In and Attend Away trials were analyzed independently. For each attention condition, only data from trials where the animal successfully detected the orientation change or from catch trials where the animal maintained fixation were used.

**Analysis: behavioral performance.** We fit the behavioral data with a logistic function and defined the threshold condition as the orientation change that was closest to the 50% threshold of the fitted psychometric function for that session. This subset of trials from within the attend-in condition in which the animal was equally likely to correctly detect (Hit) or fail to detect (Miss) the orientation change was used for our analysis. Data from Hit and Miss trials were analyzed independently.

**Preprocessing for MTwDBN analysis.** Single units were grouped according to neocortical layer (superficial/input/deep) for layer-wise (3 populations) analyses and additionally by spike waveform (narrow/broad) for layer+class (3×2 populations) analyses. The multi-unit spiking activity of these populations was used for the analysis. The data from each stimulus presentation (60–260 ms after stimulus onset) were discretized into 15 ms bins and 6 lags to give 7 time slices (-90, -75,

-60, -45, -30, -15, 0 ms). The spiking activity of each population in each bin was discretized to 1 or 0 to denote if there were spikes or not. The data from all stimulus presentations in a session/condition combination were concatenated together. Data tables had 21 columns for layerwise analyses (3 layers x 7 time slices) and 42 columns for layer-class analyses (6 layer-class populations x 7 time slices). The binned and discretized spiking activity of a single population in a single time slice was viewed as a variable in either the pairwise regression or DBN framework. Data from each session/condition were preprocessed separately. To keep analyses consistent across conditions being compared (Attend-In vs. Away OR Hit vs. Miss), the size of the bootstraps was equal to the maximum number of rows of the two conditions.

**Estimation of condition modulated edges.** Binned and sliced data tables from each session/condition were independently passed through the MTwDBN pipeline ($B = 200$) to generate 100 weighted DAGs each. Estimation of condition modulated edges was only performed for those that survived time shuffled in at least one of the conditions to be compared (Attend In or Away, Hit or Miss). For each session where both cause and effect populations were recorded, 5000 pairs of weighted DAGs were bootstrapped from the two conditions to be compared. For each pair, attention modulation indices (AMI) and hit modulation indices (HMI) were calculated as:

$$AMI = \frac{(Edge\,Weight)_{IN} - (Edge\,Weight)_{AWAY}}{(Edge\,Weight)_{IN} + (Edge\,Weight)_{AWAY}} \quad (5)$$

$$HMI = \frac{(Edge\,Weight)_{HIT} - (Edge\,Weight)_{MISS}}{(Edge\,Weight)_{HIT} + (Edge\,Weight)_{MISS}} \quad (6)$$

This resulted in 5000 modulation values for each edge that survived time-shuffling and each recording session where both cause and effect populations were recorded.

**Pairwise dependency analysis.** Total dependency weights between neocortical populations were estimated using univariate logistic regression. This analysis was only performed on edges that survived time-shuffling in the MTwDBN analysis. For each session where both cause and effect populations were recorded, 5000 samples were bootstrapped from the binned-and-sliced data tables (see *Data Preprocessing*) for each condition separately. To keep analyses consistent across Attend-In/Away and Hit/Miss, the size of the bootstraps was equal to the maximum number of rows of the two conditions to be compared. Logistic regression was performed separately from each cause variable to an effect variable using the statsmodel python package, BFGS solver[73]. The absolute value of the $\beta_1$ coefficient was treated as the total dependency weight to mirror the unsigned dependencies discovered using DBNs. This resulted in 5000 pairs of $\beta_1$ across the conditions to be compared. For each pair, attention modulation indices (AMI) and hit modulation indices (HMI) were calculated as:

$$AMI = \frac{|\beta_1^{in}| - |\beta_1^{away}|}{|\beta_1^{in}| + |\beta_1^{away}|} \quad (7)$$

$$HMI = \frac{|\beta_1^{hit}| - |\beta_1^{miss}|}{|\beta_1^{hit}| + |\beta_1^{miss}|} \quad (8)$$

This resulted in 5000 modulation values for each edge that survived time-shuffling in the MTwDBN analysis and each recording session where both cause and effect populations were recorded.

**Calculating confidence intervals of modulation indices.** Modulation indices (either attention or hit) were grouped according to time lag and additionally whether they were between layers (all layers or input ⇔ superficial only) or within layers (all layers, input only, superficial only). For layer-class analyses, they could be additionally grouped according to broad or narrow waveforms. Due to the large sample sizes generated from bootstrapping, hypothesis testing discovers all modulations to be significantly different from zero ($p = 0$). Therefore, an estimation statistics approach was used to estimate confidence intervals of modulation indices. The mean modulation index and bias-corrected and accelerated bootstrap 95% confidence intervals were calculated in python (scipy.stats.bootstrap, modified to allow for setting size of bootstrap). To estimate confidence intervals more conservatively given the large sample sizes, the number of resamples and the size of each bootstrap were each set to 5000.

**Generalized phase estimation.** To determine the temporal relationship between discrete temporal events and the ongoing local field potential (LFP) signal, we adopted the analytical signal approach for non-stationary wideband signals[86,87]. We first bandpass filtered the LFP from 5–40 Hz. This wideband signal captures the dominant fluctuations of the raw LFP signal and at the same time excludes slow global changes and higher frequency signals that could potentially contaminate the LFP. We then calculated the instantaneous phase of this wideband LFP signal (the generalized phase, GP) and studied the relationship of GP for discrete temporal events of interest (spike times, target stimulus onset time) across experimental conditions. We added a 75 ms offset to the target stimulus onset time to account for the average response latency of input layer units in our dataset[88].

**Reporting summary**
Further information on research design is available in the Nature Portfolio Reporting Summary linked to this article.

## Data availability
All binned-and-sliced data tables generated in this study have been deposited in Dryad[89] (https://doi.org/10.5061/dryad.ffbg79d2w). Raw electrophysiology data is available upon request. Source data are provided with this paper.

## Code availability
All major analyses were performed using publicly available Python packages as detailed in the methods (pgmpy, statsmodel, scipy, dit). Specific code is available from the authors upon request.

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

## Acknowledgements

This research was supported by NIH R01 EY034605, NIH R00 EY025026 to MPJ, NIH R01 EY032555, NARSAD Young Investigator Grant, Ziegler Foundation Grant, and Yale Orthwein Scholar Funds to ASN, NIH R21 MH126072 and SFARI 875855 to MPJ & ASN, NSF GRFP fellowship to AGS and by NEI core grant for vision research P30 EY026878 to Yale University. We thank Steve Chang and Weikang Shi for helpful comments on the manuscript.

## Author contributions

AD and MPJ conceptualized the project. ASN collected the electrophysiological data. AD and MPJ generated synthetic data. AGS and AD designed the data analysis pipeline. MPJ supervised the project. AGS, AD, ASN and MPJ wrote the manuscript.

## Competing interests

The authors declare no competing interests.

## Inclusion & ethics statement

We support inclusive, diverse, and equitable conduct of research.
