## [Peer Review File · Nature Communications]

Brain-state mediated modulation of inter-laminar dependencies in visual cortexREVIEWER COMMENTS

Reviewer #1 (Remarks to the Author):

Review of Das et al., (submitted to Nature Communications)

This paper's stated aim is to investigate the neural mechanisms by which spatial attention aids hierarchical computations during object processing and recognition. Specifically, they contrast 2 mechanisms proposed by prior studies; 1) increase in the efficacy of unique information directed from one encoding stage to another, and 2) an improvement in the sensory information capacity of an encoding stage through a reduction in shared fluctuations. They note that pairwise analyses have the limitation that they capture both unique and shared components of fluctuations, and thus do not differentiate between mechanisms. To test these proposals, the authors estimated attentional modulation of unique information flow across and shared information within the stages of the visual hierarchy in using the layers 4 and the superficial layers of macaque V4 as a proxy for stages. They used network-based statistical modeling to measure statistical dependencies indexing how middle and superficial cortical layers uniquely drive each other's neural activity. They found that attention strengthened unique dependencies between the input and superficial layers and then used a partial information decomposition framework to estimate modulation of shared dependencies. This suggested that within-layer shared dependencies are reduced in broad spiking (~excitatory) populations, as well as an unheralded within-layer strengthening of unique dependencies. They then examined these modulation patterns across epochs of time in which hits/misses were more/less likely. Using their findings along with earlier theoretical propositions they propose a computational model applicable to attentional enhancement and endogenous fluctuations in task performance: "enhanced information flow between and improved information capacity within encoding stages."

On the plus side, there is a lot to like about this paper. The deployment of a set of statistical approaches that can differentiate between shared and unique dependences is a novel feature and a major strength of the paper. The first author's in-depth familiarity with the data set gained through participation in the original data collection experiments is also enormously helpful to the paper. The informed application of computational modeling to index the analytic approach to some ground truth is another of the paper's strengths. Overall, the paper has the potential to be an excellent-outstanding contribution.

On the minus side, there are a few concerns, one quite broad, and several more specific, that the authors might address. The broad concern is mainly conceptual and has a number of facets. To begin with I'll identify myself (Charlie Schroeder).

BROAD CONCERN:

There is a key division of perspective in the literature on attention in non-human primates that shows up here in various forms but is not directly addressed by the authors. On the one hand, starting with the seminal studies in mid 1980s by Desimone and colleagues (including Reynolds et al) and continuing with the studies of Maunsell and colleagues, investigators have taken the perspective that an essential component of attending is suppression of low frequency excitability fluctuations (aka oscillations) and possibly enhancement of higher frequency (e.g., gamma). On the other hand, there is gathering recognition that the lower frequencies persist during attentive states and are mechanistic instruments in attentional modulation of neuronal activity in the visual system. Relevant studies in both NHPs and humans include those of the Kastner group and the Knight group (cited here, refs 15-17) concerning the role of alpha/theta range oscillations in spatial channel switching during sustained attention), and other studies by Lakatos and colleagues (and others not cited here) showing that attention uses low frequency (theta-delta range) entrainment to aid in selection of task relevant input at the expense of distractor input. The reason that this is important is that findings under the “low frequency suppression” framework generally entail random presentation of stimuli, while those in the “low frequency entrainment” framework perspective entail rhythmic or quasi-rhythmic input presentation. Reynolds and colleagues (from whose lab the empirical data for this study were drawn) make a strong case (as may be obvious in the foregoing discussion) that low frequency dynamics are more “broad-band” than “narrow-band” in character. I would agree with that.

Are these considerations relevant here? I would say yes for several reasons. First, it is likely that attention in the real-world alternates between random and rhythmic (and perhaps mixed) modes of operation (e.g., Schroeder & Lakatos, 2009; Schroeder et al., 2010), particularly since real world vision requires active saccadic sampling of information (Barczak et al., 2019); this is arguably the overt (motor) equivalent of covert spatial attention. Saccades occur quasi-rhythmically at rates of 2-5 Hz, which strongly entrains visual pathway activity at these (delta-theta) rates. Second, enhancement of information flow across layers and across areas likely requires synchronization of activity at whatever fluctuation/oscillation rate is dominant. Third, the very presence of fluctuations/oscillations, however bursty/transient and on whatever time scale (Neymotin et al., 2022; Tal et al., 2022) raises a conundrum that the authors would need to address if their proposed computational model is to have general relevance for the mechanistic study of attention: the high excitability phase is where action potential most often occur, and synchronisation of high excitability states x layers and/or areas is one of the more obvious ways to enhance information flow, HOWEVER, the high excitability phase is where one would expect “noise correlations” to be the strongest. This could mean, for e.g., that the low excitability phase is paradoxically the best state for transmission of “unique” information (see point 4 below). In any case, I am missing in this paper is a consideration of the impact of neuronal dynamics, particularly in the lower (<30 Hz) frequencies. Given that as mentioned above, real world visual processing (in monkeys and humans) usually entails moving the eyes a lot, with the attendant impact on neuronal dynamics, failure to fully consider dynamics and time may limit the generality of this otherwise fine work (again, see point 4).

MORE SPECIFIC CONCERNS:

- 1) It would be really helpful to define “unique information” and “unique dependencies” early in the paper.
- 2) The term “fluctuations” seems to correspond to neuronal “dynamics” or “oscillations” but it would be better to clear this up early rather than leaving the reader puzzling over this.
- 3) p2, top: do “laminar” stages (here middle to upper layers of V4) really represent “cross-areal” stages? I’m prepared to buy that for the moment, but one problem here that laminar projection patterns from Layer 4 to Layers 2/3 are biased in favor of the “feedforward” (layer 4 → 3) projection, whereas corticocortical projections tend more towards symmetry. It seems that the analyses here assume a ~ symmetrical 4 → 2/3 projection pattern. BTW, findings in Lakatos et al., 2008 suggest that pro- and counter-entrainment may provide a mechanism for attentional enhancement and suppression of input transmission from V1 layer 4 → 3
- 4) There is a fascinating statement on p4: “At longer timescales (> 60 ms lag)³³, while attention weakened pairwise dependencies, in agreement with previous findings^{38,39}, we found a strengthening of unique dependencies.” It is hard to extrapolate too far from this, but events separated by 60 ms could fall into opposite phases of an 8 Hz oscillation. It would be nice to see some follow-up quantification of strength of effect as a function of lag. While it is likely beyond the scope of the present analysis, hopefully the authors will be able to incorporate analysis of field potential dynamics at some point.
- 5) p6: “these behavioral fluctuations are thought to arise from endogenous fluctuations such as attentional sampling and arousal changes.” Lakatos et al., (Nat. Neurosci. 2016) provide a nice mechanistic analysis of how state fluctuations relate to input processing and behavior.
- 6) given how important accurate layer assignment is for this paper, a clearer justification of the use of the CSD profile to assign layers would be helpful. I looked at the Nandy et al paper, and that was not very helpful on this point.

Reviewer #2 (Remarks to the Author):

This is an important topic conveyed by a good (however long) abstract. I have a few comments:

- The introduction is too short, essentially only the first sentence. There should be a whole introduction on the topic before delving into details, describing the literature, competing hypotheses, the importance of the question for general topics in neuroscience, etc. Right now, the article was written for a different, more specialized journal with another format.
- The article is too short and lacks some explanations and intuitions for nearly everything. It's difficult and unpleasant to read for a neuroscientist who does not work directly on this question. It needs much work to correct this. We should be able to understand most of the article without looking at the Methods, with some intuitive explanations. For example, can you give some intuition about what is a "Dynamic Bayesian Network" and why it is an unbiased approach? Can you explain the results instead of just mentioning them and referring to the figures? Even if the article is a short communication, it should be more accessible to the broader scientific community.
- Line 78, where is the task description in the paper's main text? And of the behavior, etc.?
- The abbreviation "GLM" is mentioned as is without first explained. There is a lot of jargon specific to this topic that should be quickly explained.
- Where is the computational model mentioned in the abstract?

Reviewer #3 (Remarks to the Author):

The paper makes a great observation that responses within V4 are modulated by attention in a way that reduces dependencies selectively. There are two types of dependencies between neurons. First are dependencies that are shared, possibly due to a common source, by these neurons. Another that is information that a neuron contains that is unique to it and not present in any other neuron. They show that attention reduces the shared information reducing correlations and improves the unique information neuron representations which would improve object specific encoding.

This interesting finding would be much better served if the authors consider the following issues with the manuscript: presentation, making concepts more explicit, and describing their experimental techniques in more detail. I am aware that the short format might constrain these issues, but, currently the paper is unclear at a first reading, and to fully realize the potential of the findings it would be important for them to address them. The main issues in more detail below.

1. The first issue is providing sufficient background on a couple of fronts. On the neuroscience front, they should be more explicit about what is known about how attention can modulate neuronal responses, and how this modulation is helpful for function. For instance, they mention object recognition in the abstract. Perhaps mention the implications in discussion. Some context would then provide a better

grounding for the whole paper. Secondly, they should be more explicit about the two techniques that they used and in which part. They talk about PID as a motivation to understand unique and shared dependencies between source and target and also DBN as a way to decipher information flow. They should give some background about the use of these two techniques in neuroscience and describe briefly how it has helped.

2. The second issue relates to the first. They used a synthetic network to apply their techniques and demonstrate their applicability. They should describe in more detail where they used the PID technique and where they used the DBN technique. One has to sift through the methods and the figures to understand what they did.

3. An important methodological issue that is not clear to me is what the limitations of these techniques are. Are there some instances where they fail to pick out the information flow or where shared and unique dependencies are not deciphered? The synthetic model presents an instance where it works. This point would be moot if the synthetic network is a replica of the real one, but, it is not so clear to me that it is as I think the cortical network is more complex (related point 5 below). Perhaps, the authors can show that this is a phenomenological model that in some limit captures the essentials of the V4 network, and other connections are not central to their conclusion. Or, perhaps, PID and DBN are guaranteed to reveal the proper characteristics of any network. They should in some way show that their techniques are viable, since the paper hinges on their techniques revealing information flow and dependencies.

4. The figures are a little confusing. For instance, consider Figure 2. You have two paths in A, from re-sample and shuffle. It is not clear from the figure what is happening. And, there is very little explanation in the legends. Again, you have to dig through the methods to get some context and understand. The figures should provide a stand-alone explanation. Next, looking at Figure 3, at least to me, it wasn't clear what the bars were in e-k. Could you provide a better explanation in legends. Also, some more explanation on these results panels would also be helpful. Also, the positioning of the panels is not intuitive. I understand you are trying to conserve space, but consider making it so that you can go from left to right or top to bottom. Not, left to right, and then switching to top to bottom.

5. One issue is with some of the explanations in the methods section. They mention a synthetic network and postulate different models with 2 neurons or 6 neurons. What is the reason behind their choosing specific configurations? Is this likely to reflect the cortical network? If so, how will it scale with the actual numbers in cortex?

6. More details in other places would also be helpful. What is the precise region that was recorded from? How many neurons? Details of animals: two as mentioned in methods. These details should be at least briefly mentioned in the main text.

7. They make the point of unique and shared dependencies. How will modulation by attention help with function(what they observe)? Some thoughts on this in the discussion section of the paper would give better context to the findings.

8. They make the point that they cannot comment on the connectivity, which is understandable. But, they talk about input and superficial layers. Could they talk about connectivity between these layers, and the inputs to the region and the outputs, and how the reduction in

correlations would influence the output of V4? This goes more with providing more context for people who are not in this exact area and improving readability for the audience.

9. Minor typos like line 60, capturing "the" unique dependency structure.

10. Line 100 and 124 instead of Same can state what it is again.

Last point to consider is that a more compact abstract would pack more punch.

Reviewer #4 (Remarks to the Author):

The key findings of this paper can be summarized as follows:

The authors employed a technique called Partial Information Decomposition to distinguish various interactions both within and between regions in the visual cortex. This approach is crucial because it helps the authors disambiguate between redundant, synergistic, union, and unique components of information across and within layers of the ventral visual hierarchy. It helps them address two hypotheses, namely, that spatial attention (i) aids hierarchical computations by enhancing the transfer of unique information across layers and/or (ii) decreases the redundancy of within layer information. Other methods don't effectively isolate different components.

The authors introduced a new method termed MTwDBN for identifying dependencies between nodes in a graph. This method is a modification of Dynamic Bayesian Networks (DBN). Here, the statistical

significance of edge weights is determined using a time-shuffled estimate. The authors assert that MTwDBN outperforms existing methods, especially when dealing with sparse data.

Using the above-mentioned techniques, the authors demonstrated the following outcomes:

Enhancement of unique information across different layers

Reduction of redundant information within layers.

Increase of unique information within layers.

Notably, this increase is observed in projection neurons but not in inhibitory interneurons.

These findings are both promising and original. The paper highlights that the application of the techniques developed here was essential in reaching these conclusions. However, one critique of the paper pertains to the concise explanations provided for findings 1 and 2. Although PID has been utilized in a different context, and MTwDBN represents an extension of conventional Dynamic Bayesian methods, these aspects receive limited attention in the main body of the paper despite it being germane to the results that follow. Much of the detailed information is relegated to the supplementary methods section. While the final results (3a-d) are novel and engaging, a more comprehensive explanation of points 1 and 2 within the main paper would be appreciated. Further, there is also no mention of the caveats associated with using PID. This would be a valuable addition to the paper.

Response to Reviewers

We greatly appreciate the insightful comments and advice for improvements from all four reviewers. Below we provide detailed responses to each of the reviewers' comments and outline new analyses that are incorporated into a significantly revised and expanded manuscript. We hope that with this comprehensive revision, the editor and reviewers will find our manuscript suitable for publication at Nature Communications.

Reviewer #1 (Remarks to the Author):

Review of Das et al., (submitted to Nature Communications)

This paper's stated aim is to investigate the neural mechanisms by which spatial attention aids hierarchical computations during object processing and recognition. Specifically, they contrast 2 mechanisms proposed by prior studies; 1) increase in the efficacy of unique information directed from one encoding stage to another, and 2) an improvement in the sensory information capacity of an encoding stage through a reduction in shared fluctuations. They note that pairwise analyses have the limitation that they capture both unique and shared components of fluctuations, and thus do not differentiate between mechanisms. To test these proposals, the authors estimated attentional modulation of unique information flow across and shared information within the stages of the visual hierarchy in using the layers 4 and the superficial layers of macaque V4 as a proxy for stages. They used network-based statistical modeling to measure statistical dependencies indexing how middle and superficial cortical layers uniquely drive each other's neural activity. They found that attention strengthened unique dependencies between the input and superficial layers and then used a partial information decomposition framework to estimate modulation of shared dependencies. This suggested that within-layer shared dependencies are reduced in broad spiking (~excitatory) populations, as well as an unheralded within-layer strengthening of unique dependencies. They then examined these modulation patterns across epochs of time in which hits/misses were more/less likely. Using their findings along with earlier theoretical propositions they propose a computational model applicable to attentional enhancement and endogenous fluctuations in task performance: "enhanced information flow between and improved information capacity within encoding stages."

On the plus side, there is a lot to like about this paper. The deployment of a set of statistical approaches that can differentiate between shared and unique dependences is a novel feature and a major strength of the paper. The first author's in-depth familiarity with the data set gained through participation in the original data collection experiments is also enormously helpful to the paper. The informed application of computational modeling to index the analytic approach to some ground truth is another of the paper's strengths. Overall, the paper has the potential to be an excellent-outstanding contribution.

We thank the reviewer for their enthusiastic assessment of our study.

On the minus side, there are a few concerns, one quite broad, and several more specific, that the authors might address. The broad concern is mainly conceptual and has a number of facets. To begin with I'll identify myself (Charlie Schroeder).

We appreciate the detailed and constructive feedback provided by Dr. Schroeder. It has not only improved our manuscript, but the novel analysis inspired by the feedback has shed new light on potential mechanisms of enhanced inter-laminar information flow.

BROAD CONCERN:

There is a key division of perspective in the literature on attention in non-human primates that shows up here in various forms but is not directly addressed by the authors. On the one hand, starting with the seminal studies in mid 1980s by Desimone and colleagues (including Reynolds et al) and continuing with the studies of Maunsell and colleagues, investigators have taken the perspective that an essential component of attending is suppression of low frequency excitability fluctuations (aka oscillations) and possibly enhancement of higher frequency (e.g., gamma). On the other hand, there is gathering recognition that the lower frequencies persist during attentive states and are mechanistic instruments in attentional modulation of neuronal activity in the visual system. Relevant studies in both NHPs and humans include those of the Kastner group and the Knight group (cited here, refs 15-17) concerning the role of alpha/theta range oscillations in spatial channel switching during sustained attention), and other studies by Lakatos and colleagues (and others not cited here) showing that attention uses low frequency (theta-delta range) entrainment to aid in selection of task relevant input at the expense of distractor input. The reason that this is important is that findings under the “low frequency suppression” framework generally entail random presentation of stimuli, while those in the “low frequency entrainment” framework perspective entail rhythmic or quasi-rhythmic input presentation. Reynolds and colleagues (from whose lab the empirical data for this study were drawn) make a strong case (as may be obvious in the foregoing discussion) that low frequency dynamics are more “broad-band” than “narrow-band” in character. I would agree with that.

Are these considerations relevant here? I would say yes for several reasons. First, it is likely that attention in the real-world alternates between random and rhythmic (and perhaps mixed) modes of operation (e.g., Schroeder & Lakatos, 2009; Schroeder et al., 2010), particularly since real world vision requires active saccadic sampling of information (Barczak et al., 2019); this is arguably the overt (motor) equivalent of covert spatial attention. Saccades occur quasi-rhythmically at rates of 2-5 Hz, which strongly entrains visual pathway activity at these (delta-theta) rates. Second, enhancement of information flow across layers and across areas likely requires synchronization of activity at whatever fluctuation/oscillation rate is dominant. Third, the very presence of fluctuations/oscillations, however bursty/transient and on whatever time scale (Neymotin et al., 2022; Tal et al., 2022) raises a conundrum that the authors would need to address if their proposed computational model is to have general relevance for the mechanistic study of attention: the high excitability phase is where action potential most often occur, and synchronisation of high excitability states x layers and/or areas is one of the more obvious ways to enhance information flow, HOWEVER, the high excitability phase is where one would expect “noise correlations” to be the strongest. This could mean, for e.g., that the low excitability phase is paradoxically the best state for transmission of “unique” information (see point 4 below). In any case, I am missing in this paper is a consideration of the impact of neuronal dynamics, particularly in the lower (<30 Hz) frequencies. Given that as mentioned above, real world visual processing (in monkeys and humans) usually entails moving the eyes a lot, with the attendant impact on neuronal dynamics, failure to fully consider dynamics and time may limit the generality of this otherwise fine work (again, see point 4).

Dr. Schroeder has raised several excellent points here regarding potential mechanisms of attentional enhancement in the context of inter-stage communication within the hierarchical architecture of the ventral visual pathway. Especially helpful has been the hypothesis he has articulated underlying the neural mechanism of unique dependency enhancement between the input and superficial layers: namely, that the best state for the transmission of “unique” information could be the phases of low neural

excitability. To directly test this hypothesis, we conducted new analyses that we now report in our Results (In 214-227; Figs 5f,g and S5):

“To test if optimal states that are associated with hit trials are also associated with a reduction in shared correlations among the projection neurons of the input layer, we estimated the probability of presentation of “successful” target stimuli and the probability of spiking of input layer broad spiking units, both as a function of the phase of the ongoing cortical activity. We estimated the generalized phase of the band-filtered (5-40 Hz) local field potential signals in the input layer (see Methods), and calculated the probability of a ‘hit’-causing target onset and of neuronal spikes at different phases (Fig 5f). We found a clear phase dependence of response onset of ‘hit’ targets (Fig 5g, top). These phases were also associated with a lower excitability of broad spiking cells (Fig 5g, bottom), suggesting that the improved performance in optimal states occurs during phases of lower than average spiking probability of putative excitatory neurons in the input layer. Interestingly, the excitability of superficial layer putative excitatory neurons, the primary candidates that project to downstream cortical areas in the ventral stream, was independent of the phase of the ongoing activity in the superficial layers of V4 (Fig S5).”

We have added a discussion of the implications of these findings within the context of the above hypothesis (In 266-274):

“Additionally, we find that phases of the endogenous fluctuations that are associated with optimal target presentation (resulting in hits) are also associated with reduced excitability of broad spiking neurons, especially in the input layer. Our findings are in agreement with previous reports of rhythmic shifts of neural excitability and their entrainment to the stream of sensory inputs as key mechanisms of sensory selection¹⁻³. Interpreting the fluctuations in excitability to be at least partly based on changing correlations due to fluctuations in shared inputs, our finding suggests an additional mechanism through which weakened shared neural activity fluctuations could improve behavioral outcome: a lowered excitability, which is associated with reliable encoding in the visual cortex⁴”

We thank Dr. Schroeder again for this excellent suggestion which we believe has complemented our causal model by providing a concrete mechanism of the transmission of unique information across cortical layers.

MORE SPECIFIC CONCERNS:

1) It would be really helpful to define “unique information” and “unique dependencies” early in the paper.

We agree and have incorporated this suggestion in our rewritten manuscript. Please see the updated Results section (In 82-90) and the corresponding revised figures (Fig 1a, Fig S1).

2) The term “fluctuations” seems to correspond to neuronal “dynamics” or “oscillations” but it would be better to clear this up early rather than leaving the reader puzzling over this.

We agree that it is necessary to qualify “fluctuations” to provide clarity in the text. The term was previously used to describe the dynamics of neural activity, behavioral performance, or brain state in different parts of the manuscript. We have now made the necessary corrections in the revised manuscript.

3) p2, top: do “laminar” stages (here middle to upper layers of V4) really represent “cross-areal” stages? I’m prepared to buy that for the moment, but one problem here that laminar projection patterns from Layer 4 to Layers 2/3 are biased in favor of the “feedforward” (layer 4 \rightarrow 3) projection, whereas corticocortical projections tend more towards symmetry. It seems that the analyses here assume a \sim symmetrical 4 \leftrightarrow 2/3 projection pattern. BTW, findings in Lakatos et al., 2008 suggest that pro- and counter-entrainment may provide a mechanism for attentional enhancement and suppression of input transmission from V1 layer 4 \rightarrow 3

We agree that there are differences in the inter-laminar vs inter-areal connectivity when considering the balance of feedforward/feedback projections. The aspect of hierarchical architecture that we focus on in this study is a robust feedforward connectivity motif – present in both inter-laminar and inter-areal circuits – that supports the downstream information flow crucial for object recognition. However, the point raised by Dr. Schroeder is an important one and will be a focus of future queries that we discuss in the revision (how are dependency components modulated in feedforward vs. feedback direction? In 276-289). The agreement of our findings (in a feedforward-dominated circuit) with previous studies showing enhanced communication through both correlation-based analyses and causal manipulation (electrical stimulation) in the more balanced V1->MT Dorsal pathway⁵ suggests that it would be reasonable to assume similar motifs of attentional modulation at both inter-laminar and inter-areal levels.

4) There is a fascinating statement on p4: “At longer timescales (> 60 ms lag)³³, while attention weakened pairwise dependencies, in agreement with previous findings^{38,39}, we found a strengthening of unique dependencies.” It is hard to extrapolate too far from this, but events separated by 60 ms could fall into opposite phases of an 8 Hz oscillation. It would be nice to see some follow-up quantification of strength of effect as a function of lag. While it is likely beyond the scope of the present analysis, hopefully the authors will be able to incorporate analysis of field potential dynamics at some point.

We have heeded this suggestion and have added new results regarding field potential. Since we find strengthening of dependencies at multiple lags, we analyzed a broadband 5-40 Hz signal to examine the relative phase relationship in a more unbiased manner. Our new analysis (Fig. 5f-g) of the broadband LFP signal does suggest optimal phases of stimulus presentation that also correlate with phases of low excitability, in a cell-class specific way) (In 213-227). Please also see our response to the main point above.

5) p6: “these behavioral fluctuations are thought to arise from endogenous fluctuations such as attentional sampling and arousal changes.” Lakatos et al., (Nat. Neurosci. 2016) provide a nice mechanistic analysis of how state fluctuations relate to input processing and behavior.

We thank Dr. Schroeder for bringing this pertinent study to our attention. We now discuss it in our manuscript when we interpret our new result (Fig 5 f,g) in the rewritten Discussion section (In (266-274).

6) given how important accurate layer assignment is for this paper, a clearer justification of the use of the CSD profile to assign layers would be helpful. I looked at the Nandy et al paper, and that was not very helpful on this point.

We have now added text in the Methods section to elaborate on the criteria used for assigning layer boundaries (In 762-769). We have also added an example CSD profile marked up to illustrate our

approach in a new main figure (Fig. 3).

Reviewer #2 (Remarks to the Author):

This is an important topic conveyed by a good (however long) abstract.

We thank the reviewer for enthusiastically supporting our manuscript.

I have a few comments:

- The introduction is too short, essentially only the first sentence. There should be a whole introduction on the topic before delving into details, describing the literature, competing hypotheses, the importance of the question for general topics in neuroscience, etc. Right now, the article was written for a different, more specialized journal with another format.

We completely agree with this assessment and have now extensively rewritten and expanded the manuscript.

- The article is too short and lacks some explanations and intuitions for nearly everything. It's difficult and unpleasant to read for a neuroscientist who does not work directly on this question. It needs much work to correct this. We should be able to understand most of the article without looking at the Methods, with some intuitive explanations. For example, can you give some intuition about what is a "Dynamic Bayesian Network" and why it is an unbiased approach? Can you explain the results instead of just mentioning them and referring to the figures? Even if the article is a short communication, it should be more accessible to the broader scientific community.

We have amended these shortcomings by completely rewriting and expanding the manuscript. We now summarize our approach in the Results section (ln 111-121) and provide intuitions for the concepts with the help of the redesigned DBN schematics in the related figure (Fig. 2). Our expanded Results section elaborates on each of our findings. Further, we have elaborated on the implications of the Results, specifically the functional role of the modulation of different components of dependencies in optimal sensory states, in the revised Discussion section.

- Line 78, where is the task description in the paper's main text? And of the behavior, etc.?

We have updated the Results section (ln 141-156) and added a new main figure (Fig. 3) to address this point. We have also expanded the Methods section to provide further details (ln 737-753).

- The abbreviation "GLM" is mentioned as is without first explained. There is a lot of jargon specific to this topic that should be quickly explained.

We have corrected this and added text to summarize the GLM approach (ln 312-323).

- Where is the computational model mentioned in the abstract?

We regret the confusion in the wording. We have rewritten our abstract and now clearly refer to a conceptual model where needed in the manuscript.

Reviewer #3 (Remarks to the Author):

The paper makes a great observation that responses within V4 are modulated by attention in a way that reduces dependencies selectively. There are two types of dependencies between neurons. First are dependencies that are shared, possibly due to a common source, by these neurons. Another that is information that a neuron contains that is unique to it and not present in any other neuron. They show that attention reduces the shared information reducing correlations and improves the unique information neuron representations which would improve object specific encoding.

We appreciate the enthusiastic review of our study and the detailed suggestions for improvement of the manuscript.

This interesting finding would be much better served if the authors consider the following issues with the manuscript: presentation, making concepts more explicit, and describing their experimental techniques in more detail. I am aware that the short format might constrain these issues, but, currently the paper is unclear at a first reading, and to fully realize the potential of the findings it would be important for them to address them. The main issues in more detail below.

1. The first issue is providing sufficient background on a couple of fronts. On the neuroscience front, they should be more explicit about what is known about how attention can modulate neuronal responses, and how this modulation is helpful for function. For instance, they mention object recognition in the abstract. Perhaps mention the implications in discussion. Some context would then provide a better grounding for the whole paper.

Our rewritten manuscript now addresses this by providing a background in the Introduction (ln 47-58) and implications of the results in the Discussion section (ln 231-274).

Secondly, they should be more explicit about the two techniques that they used and in which part. They talk about PID as a motivation to understand unique and shared dependencies between source and target and also DBN as a way to decipher information flow. They should give some background about the use of these two techniques in neuroscience and describe briefly how it has helped.

We reworked Figures 1 and 2 to break down the build-up of the concepts underlying our techniques. We also summarize our techniques in the Results section (ln 82-88, 111-125) and elaborate on them in the Methods section of the main text (ln 515-537, 572-606).

2. The second issue relates to the first. They used a synthetic network to apply their techniques and demonstrate their applicability. They should describe in more detail where they used the PID technique and where they used the DBN technique. One has to sift through the methods and the figures to understand what they did.

We have fully rewritten the main text, reworked Figures 1 and 2, and expanded the Methods to address these points.

3. An important methodological issue that is not clear to me is what the limitations of these techniques

are. Are there some instances where they fail to pick out the information flow or where shared and unique dependencies are not deciphered? The synthetic model presents an instance where it works. This point would be moot if the synthetic network is a replica of the real one, but, it is not so clear to me that it is as I think the cortical network is more complex (related point 5 below). Perhaps, the authors can show that this is a phenomenological model that in some limit captures the essentials of the V4 network, and other connections are not central to their conclusion. Or, perhaps, PID and DBN are guaranteed to reveal the proper characteristics of any network. They should in some way show that their techniques are viable, since the paper hinges on their techniques revealing information flow and dependencies.

We scope the discussion of our results as follows:

“The laminar network is considered a canonical circuit that constitutes a key computational unit in the cortex. While anatomical connectivity maps have identified the key variables of this unit^{6,7}, access to the functional connectivity that determines the resulting computations has remained elusive. Laminar recordings in awake animals transitioning across behavioral states have allowed us to observe the neural variables that are expected to play a significant role in these computations.” (In 231-236)

We further discuss the caveats of our approach in the Discussion section of the revised manuscript (In 325-347).

Additionally, the stimuli used in our experiment were of a level of complexity comparable to the tuning of V4 neurons, hence feedback from higher areas was not expected to play an important role in the neural activity in V4. The neural variable that was expected to have a key role in the local activity is the input from regions (e.g. frontal eye fields) that signal the attentive state. Since this was the independent variable that was explicitly manipulated in our experiment, we compared how this variable modulates the dependency structure in the local V4 network.

4. The figures are a little confusing. For instance, consider Figure 2. You have two paths in A, from re-sample and shuffle. It is not clear from the figure what is happening. And, there is very little explanation in the legends. Again, you have to dig through the methods to get some context and understand. The figures should provide a stand-alone explanation.

We have reorganized Figure 2, and updated its caption and expanded the Methods section (Multi-time lag Weighted Dynamic Bayesian Network (MTwDBN) Analysis Pipeline; In 572-606) to address these issues in a comprehensive manner.

Next, looking at Figure 3, at least to me, it wasn't clear what the bars were in e-k. Could you provide a better explanation in legends. Also, some more explanation on these results panels would also be helpful.

We have added helpful schematics to figure panels a/g and explanatory text in the caption for this figure (now Fig. 4). Additionally, we have rewritten the Results section related to this figure to further clarify the results shown in Fig. 4 (In 158-194).

Also, the positioning of the panels is not intuitive. I understand you are trying to conserve space, but consider making it so that you can go from left to right or top to bottom. Not, left to right, and then switching to top to bottom.

We have now fixed this issue in the newly organized figure (now Fig. 4).

5. One issue is with some of the explanations in the methods section. They mention a synthetic network and postulate different models with 2 neurons or 6 neurons. What is the reason behind their choosing specific configurations? Is this likely to reflect the cortical network? If so, how will it scale with the actual numbers in cortex?

We used the synthetic networks to investigate if (a) edge weight reflects dependency strength and (b) how well our method discovers the dependencies in a recurrent multi-variate network for which the ground truth is known. The 2-network configuration was used to investigate the relationship of "edge weight" in our MTwDBN graphs to the underlying dependency strength. The 6-population network was built to investigate how the method performs in a multi-variable network with recurrent connectivity. Since the relationship between dependency strength and synaptic connection strength in this kind of network is complex, we used two separate configurations to investigate these two points.

6. More details in other places would also be helpful. What is the precise region that was recorded from? How many neurons? Details of animals: two as mentioned in methods. These details should be at least briefly mentioned in the main text.

We summarize these details in the main text (ln 143-147). Additionally, we have moved all the experimental details to the Methods section of main text and added a new main figure (Figure 3) to illustrate the experimental and recoding paradigm.

7. They make the point of unique and shared dependencies. How will modulation by attention help with function(what they observe)? Some thoughts on this in the discussion section of the paper would give better context to the findings.

We agree and now elaborate on this in the new Discussion section (ln 231-274).

8. They make the point that they cannot comment on the connectivity, which is understandable. But, they talk about input and superficial layers. Could they talk about connectivity between these layers, and the inputs to the region and the outputs, and how the reduction in correlations would influence the output of V4? This goes more with providing more context for people who are not in this exact area and improving readability for the audience.

Point well taken. We now provide a brief summary of these points in the Introduction (ln 60-78) and extensively explore them in Discussion (ln 231-274).

9. Minor typos like line 60, capturing "the" unique dependency structure.

We fixed this (now ln 114).

10. Line 100 and 124 instead of Same can state what it is again.

We have explicitly stated the captions in the revised figure (now Fig. 4).

Last point to consider is that a more compact abstract would pack more punch.

We agree and have written a concise new abstract that highlights the key findings of our study.

Reviewer #4 (Remarks to the Author):

The key findings of this paper can be summarized as follows:

The authors employed a technique called Partial Information Decomposition to distinguish various interactions both within and between regions in the visual cortex. This approach is crucial because it helps the authors disambiguate between redundant, synergistic, union, and unique components of information across and within layers of the ventral visual hierarchy. It helps them address two hypotheses, namely, that spatial attention (i) aids hierarchical computations by enhancing the transfer of unique information across layers and/or (ii) decreases the redundancy of within layer information. Other methods don't effectively isolate different components.

The authors introduced a new method termed MTwDBN for identifying dependencies between nodes in a graph. This method is a modification of Dynamic Bayesian Networks (DBN). Here, the statistical significance of edge weights is determined using a time-shuffled estimate. The authors assert that MTwDBN outperforms existing methods, especially when dealing with sparse data.

Using the above-mentioned techniques, the authors demonstrated the following outcomes:

Enhancement of unique information across different layers

Reduction of redundant information within layers.

Increase of unique information within layers.

Notably, this increase is observed in projection neurons but not in inhibitory interneurons.

These findings are both promising and original. The paper highlights that the application of the techniques developed here was essential in reaching these conclusions.

We thank the reviewer for their enthusiastic evaluation of our study.

However, one critique of the paper pertains to the concise explanations provided for findings 1 and 2. Although PID has been utilized in a different context, and MTwDBN represents an extension of conventional Dynamic Bayesian methods, these aspects receive limited attention in the main body of the paper despite it being germane to the results that follow. Much of the detailed information is relegated to the supplementary methods section.

We agree with this critique and have now completely re-written the manuscript (including updated figures) to address this:

1. *Figure 1 has been updated to illustrate the conceptual framework (PID) used in our analysis.*

2. *Figure 2 has been updated to break down the step-by-step conceptual advancement from BN → DBN → MTwDBN.*
3. *All germane information in the supplementary text has been moved to the main body [Results (ln 80-108, 110-138) and Methods] of the paper. There is no supplementary text in the revised manuscript.*

While the final results (3a-d) are novel and engaging, a more comprehensive explanation of points 1 and 2 within the main paper would be appreciated.

We agree with this comment and have addressed it in the Results section (ln 158-194) of the revised manuscript. Additionally, we have expanded the interpretation of these results in the re-written Discussion section (ln 231-295).

Further, there is also no mention of the caveats associated with using PID. This would be a valuable addition to the paper.

We have addressed this critique by elaborating on our previously stated limitations in our rewritten Discussion section (ln 325-347).

- 1 Schroeder, C. E., Wilson, D. A., Radman, T., Scharfman, H. & Lakatos, P. Dynamics of Active Sensing and perceptual selection. *Current opinion in neurobiology* **20**, 172-176, doi:10.1016/j.conb.2010.02.010 (2010).
- 2 Schroeder, C. E. & Lakatos, P. Low-frequency neuronal oscillations as instruments of sensory selection. *Trends in neurosciences* **32**, 9-18, doi:10.1016/j.tins.2008.09.012 (2009).
- 3 Lakatos, P., Karmos, G., Mehta, A. D., Ulbert, I. & Schroeder, C. E. Entrainment of neuronal oscillations as a mechanism of attentional selection. *Science* **320**, 110-113, doi:10.1126/science.1154735 (2008).
- 4 Haider, B. *et al.* Synaptic and network mechanisms of sparse and reliable visual cortical activity during nonclassical receptive field stimulation. *Neuron* **65**, 107-121, doi:10.1016/j.neuron.2009.12.005 (2010).
- 5 Ruff, D. A. & Cohen, M. R. Attention Increases Spike Count Correlations between Visual Cortical Areas. *The Journal of neuroscience : the official journal of the Society for Neuroscience* **36**, 7523-7534, doi:10.1523/JNEUROSCI.0610-16.2016 (2016).
- 6 Douglas, R. J. & Martin, K. A. C. Recurrent neuronal circuits in the neocortex. *Current biology : CB* **17**, R496-500, doi:10.1016/j.cub.2007.04.024 (2007).
- 7 Douglas, R. J. & Martin, K. A. C. Mapping the matrix: the ways of neocortex. *Neuron* **56**, 226-238, doi:10.1016/j.neuron.2007.10.017 (2007).

REVIEWERS' COMMENTS

Reviewer #1 (Remarks to the Author):

Review of Das et al., (submitted to Nature Communications)

The authors have done an excellent job responding to my comments. I applaud their efforts. I have a few more comments the authors may wish to consider. These are suggestions for clarifying and strengthening the paper's message, but I would not insist that they be followed.

First, an interesting connection on P10: "We find that unique dependencies between input and superficial layer populations are strengthened by attention, as well as during successful behavioral outcomes within the attentive state. This finding suggests that enhanced unique information transfer between encoding stages of the laminar hierarchy is a hallmark of behaviorally optimal sensory processing." It is intriguing that this attentional enhancement of Layer 4 to supragranular "connectivity" is reminiscent of Lakatos et al., (2008) in V1; i.e., in the attend visual condition, Layer 4 and Layer 3 shared common high and low excitability phases, while in the attend auditory (ignore visual) condition, excitability in Layers 4 and 3 fluctuated in counterphase.

P11: "our finding suggests an additional mechanism through which weakened shared neural activity fluctuations could improve behavioral outcome: a lowered excitability, which is associated with reliable encoding in the visual cortex68." This might be clearer if you stated "... a low excitability phase of an endogenous fluctuation, which is associated..."

P12: "...optogenetic induction of low-frequency fluctuations in V4 impairs performance in an attention-demanding task71. An important follow-up..." This experiment tested a single circumstance - the case in which target occurrence is temporally unpredictable; this is the case where the brain tries to reduce low frequency fluctuation amplitudes (the "random mode in Schroeder & Lakatos). A more common circumstance is the case where target occurrence is temporally predictable (e.g., in natural free viewing of a scene), and the brain can use ambient rhythms rather than suppressing them (the rhythmic mode). In my opinions this would be an even more impactful follow-up, as it would broaden the applicability of the work to encompass variations in shared dependencies due to endogenous fluctuations (first sentence in your conclusion)

Again, these are suggestions. The paper is acceptable as is.

Charlie Schroeder

Reviewer #2 (Remarks to the Author):

Thank you for addressing the points I raised.

Reviewer #3 (Remarks to the Author):

The paper is much improved from the previous version, which I liked and appreciate. I would emphasize the following points in making the paper more accessible to non-vision neuroscientists and, possibly more lay people.

The abstract is better, but they could consider a further distillation of information. Also, if they could use simpler words instead of endogenous, putative and unique dependencies it would be better as these are concepts that are described further on and seem a little jargonish in the abstract.

I am still a bit unclear about points 3 (how realistic is the synthetic network) and 5 (why the 2 vs. 6 network models). Perhaps, my confusion about both points is connected. They speak about a canonical circuit. It might help in what way the synthetic network mimics the the real one. It could be as simple as 2 pyramidal cells in layer 4 connect to x neurons in the layer X and this resembles the circuit in some way. It would help picture how the network is similar and also attach how the computations occur.

I mention these points in the interest of making the paper understandable to a wider audience. I am happy with the improvements and the results as they stand. I think a simpler more understandable paper would help improve the paper's visibility and interest in it.

Reviewer #4 (Remarks to the Author):

The authors have addressed my questions convincingly.

Response to Reviewers

We greatly appreciate the insightful comments and advice from all four reviewers. Below we provide responses to each of the reviewers' comments. We hope that with this final revision, the readers at Nature Communications will find our article clear and insightful.

Reviewer #1 (Remarks to the Author):

Review of Das et al., (submitted to Nature Communications)

The authors have done an excellent job responding to my comments. I applaud their efforts. I have a few more comments the authors may wish to consider. These are suggestions for clarifying and strengthening the paper's message, but I would not insist that they be followed.

First, an interesting connection on P10: "We find that unique dependencies between input and superficial layer populations are strengthened by attention, as well as during successful behavioral outcomes within the attentive state. This finding suggests that enhanced unique information transfer between encoding stages of the laminar hierarchy is a hallmark of behaviorally optimal sensory processing." It is intriguing that this attentional enhancement of Layer 4 to supragranular "connectivity" is reminiscent of Lakatos et al., (2008) in V1; i.e., in the attend visual condition, Layer 4 and Layer 3 shared common high and low excitability phases, while in the attend auditory (ignore visual) condition, excitability in Layers 4 and 3 fluctuated in counterphase.

We have added this point to our discussion.

P11: "our finding suggests an additional mechanism through which weakened shared neural activity fluctuations could improve behavioral outcome: a lowered excitability, which is associated with reliable encoding in the visual cortex⁶⁸." This might be clearer if you stated "... a low excitability phase of an endogenous fluctuation, which is associated..."

We have incorporated this suggestion for better readability.

P12: "...optogenetic induction of low-frequency fluctuations in V4 impairs performance in an attention-demanding task⁷¹. An important follow-up..." This experiment tested a single circumstance - the case in which target occurrence is temporally unpredictable; this is the case where the brain tries to reduce low frequency fluctuation amplitudes (the "random mode in Schroeder & Lakatos). A more common circumstance is the case where target occurrence is temporally predictable (e.g., in natural free viewing of a scene), and the brain can use ambient rhythms rather than suppressing them (the rhythmic mode). In my opinions this would be an even more impactful follow-up, as it would broaden the applicability of the work to encompass variations in shared dependencies due to endogenous fluctuations (first sentence in your conclusion)

Again, these are suggestions. The paper is acceptable as is.

We have added this point to our discussion.

Charlie Schroeder

Reviewer #2 (Remarks to the Author):

Thank you for addressing the points I raised.

We appreciate all the feedback.

Reviewer #3 (Remarks to the Author):

The paper is much improved from the previous version, which I liked and appreciate. I would emphasize the following points in making the paper more accessible to non-vision neuroscientists and, possibly more lay people.

The abstract is better, but they could consider a further distillation of information. Also, if they could use simpler words instead of endogenous, putative and unique dependencies it would be better as these are concepts that are described further on and seem a little jargonish in the abstract.

We have revised and shortened the abstract for better comprehensibility.

I am still a bit unclear about points 3 (how realistic is the synthetic network) and 5 (why the 2 vs. 6 network models). Perhaps, my confusion about both points is connected. They speak about a canonical circuit. It might help in what way the synthetic network mimics the the real one. It could be as simple as 2 pyramidal cells in layer 4 connect to x neurons in the layer X and this resembles the circuit in some way. It would help picture how the network is similar and also attach how the computations occur.

We have now removed confusing text from the description of the model and state the model description as follows:

“To validate our analysis pipeline, we simulated a compartmentalized network of stochastically spiking excitatory and inhibitory neurons that were recurrently connected within compartments, in addition to inter-compartmental excitatory connectivity.”

I mention these points in the interest of making the paper understandable to a wider audience. I am happy with the improvements and the results as they stand. I think a

simpler more understandable paper would help improve the paper's visibility and interest in it.

We completely agree and appreciate all the suggestions for improving the paper.

Reviewer #4 (Remarks to the Author):

The authors have addressed my questions convincingly.

We appreciate all the feedback.